# iRhom pseudoproteases regulate ER stress-induced cell death through IP$_3$ receptors and BCL-2

Iqbal Dulloo [1✉], Peace Atakpa-Adaji[2], Yi-Chun Yeh[3], Clémence Levet[1], Sonia Muliyil[1], Fangfang Lu[1], Colin W. Taylor [2] & Matthew Freeman [1✉]

The folding capacity of membrane and secretory proteins in the endoplasmic reticulum (ER) can be challenged by physiological and pathological perturbations, causing ER stress. If unresolved, this leads to cell death. We report a role for iRhom pseudoproteases in controlling apoptosis due to persistent ER stress. Loss of iRhoms causes cells to be resistant to ER stress-induced apoptosis. iRhom1 and iRhom2 interact with IP$_3$ receptors, critical mediators of intracellular Ca$^{2+}$ signalling, and regulate ER stress-induced transport of Ca$^{2+}$ into mitochondria, a primary trigger of mitochondrial membrane depolarisation and cell death. iRhoms also bind to the anti-apoptotic regulator BCL-2, attenuating the inhibitory interaction between BCL-2 and IP$_3$ receptors, which promotes ER Ca$^{2+}$ release. The discovery of the participation of iRhoms in the control of ER stress-induced cell death further extends their potential pathological significance to include diseases dependent on protein misfolding and aggregation.

[1] Dunn School of Pathology, University of Oxford, South Parks Road, Oxford OX1 3RE, UK. [2] Department of Pharmacology, University of Cambridge, Tennis Court Road, Cambridge CB2 1PD, UK. [3] Department of Physiology, Anatomy and Genetics, University of Oxford, South Parks Road, Oxford OX1 3PT, UK. ✉email: iqbal.dulloo@path.ox.ac.uk; matthew.freeman@path.ox.ac.uk

An unexpected outcome of the genomics era has been the discovery of numerous conserved genes encoding proteins structurally related to active enzymes, but which are missing essential active site residues, rendering them catalytically dead[1,2]. It is becoming clear that these 'pseudoenzymes' are not just evolutionary remnants but have important functions in their own right, often derived from the role of their active enzyme ancestors[3]. iRhoms are catalytically inactive relatives of rhomboid intramembrane serine proteases, which are conserved in all metazoans[4,5]. Like other members of the rhomboid-like superfamily, a core molecular function of iRhoms is the specific recognition of transmembrane domains (TMDs) of interacting client proteins[6]. They have been implicated in a number of different cellular control processes but are best characterised as regulatory cofactors of the metalloprotease ADAM17, and consequently of inflammatory and growth factor signalling[2,5,7].

Although iRhoms are distributed throughout the secretory pathway, and the recent focus on regulation of ADAM17 has tended to emphasise their role at the plasma membrane[8], in most cells, overexpressed iRhoms are predominantly located in the endoplasmic reticulum (ER), although they are also detectable later in the secretory pathway and at the plasma membrane. In the ER they have reported functions in protein homoeostasis[9,10], ER to Golgi trafficking[11,12], and regulating the cellular response to viruses[10,13]. One of the central functions of the ER is the folding and maturation of secreted proteins and most transmembrane proteins[14]. Reflecting the importance of these classes of proteins, which comprise about one-third of the human genome, the ER has evolved sophisticated quality control machinery. Disturbance of the equilibrium between protein biosynthesis, folding, maturation and onward trafficking leads to potentially toxic accumulation of unfolded or misfolded proteins, creating a state of ER stress. This triggers an adaptive mechanism called the unfolded protein response (UPR), which restores ER homoeostasis and promotes cellular survival[15]. For example, the expression of folding chaperones is upregulated, while terminally unfolded proteins are retrotranslocated into the cytosol for proteasomal degradation via ER-associated degradation (ERAD)[16]. Broadly the UPR is mediated by three alternative sensing pathways[15,17], controlled respectively by ATF6 (activating transcription factor 6), IRE1 (inositol-requiring protein 1), and PERK (protein kinase RNA-like ER kinase). Prolonged or overwhelming ER stress that cannot be relieved by the adaptive UPR phase triggers apoptotic death of the faulty cell[18,19]. ER stress-induced apoptosis is regulated by a combination of several pathways but is mainly induced by PERK/eIF2α-dependent induction of the pro-apoptotic transcription factor C/EBP homologous protein (CHOP), and IRE1-mediated activation of TRAF2/ASK1/JNK/p38 kinase cascade[18,19]. Crosstalk between both pathways activates CHOP, which therefore acts as the main inducer of downstream mitochondria-mediated cell death[20,21]. Deregulation of this stress response system has been implicated in several human pathologies including metabolic syndromes, neurodegenerative diseases, and cancer[22,23]. Modulators of ER stress pathways have therefore been studied as potential targets for therapeutic intervention.

There have been reports indicating that, in addition to their other functions, iRhoms have a role in processes associated with ER stress responses. The first of these demonstrated that *Drosophila* iRhom can trigger the degradation of EGF-ligands in a process that resembled ERAD[9]. In another case, it was reported that iRhom1 is a regulator of proteasome activity under ER stress conditions in both human cells and flies[24]. Absence of iRhom1 prevents the dimerisation of proteasome assembly chaperone 1 and 2 (PAC1 and PAC2), leading to impaired function of the 26S proteasome complex[24]. More recently, it was proposed that a high-fat diet leads to ER stress-associated induction of iRhom2 in mice, promoting cardiomyocyte inflammation via the iRhom2/ADAM17 pathway, and lipid deposition in the heart[25]. Together, these studies suggest a functional link between ER stress responses and iRhoms, although there are few details of either the cellular mechanisms or the physiological significance.

Here we show that iRhom1 and iRhom2 have an essential role in the response to chronic ER stress. In mammals and *Drosophila*, the absence of iRhoms inhibits ER-stress induced cell death. The adaptive UPR is normal in cells deficient for iRhoms but under persistent ER stress, cell death is inhibited. Our data show that iRhoms are needed for the activation of apoptotic caspases, specifically via the intrinsic mitochondrial cell death pathway. We also show that the ability of inositol 1,4,5-trisphosphate receptors (IP$_3$Rs) to release ER Ca$^{2+}$—a major signal that initiates the mitochondrial death pathway—is dependent on the function of iRhoms. At a molecular level, iRhoms bind to both IP$_3$Rs and their negative regulator BCL-2, modulating their interaction. Overall, we identify a previously unappreciated role of iRhoms in regulating ER stress-induced death, which involves regulation of the inhibitory effect of anti-apoptotic BCL-2 on the function of IP$_3$Rs.

## Results

**ER stress-induced cell death is defective in the absence of iRhoms.** To explore the role of iRhoms in the ER stress response, we investigated how cells deficient for iRhoms behave in the presence of ER stressors. iRhom1 and iRhom2 have high sequence similarity and overlapping functions so we examined the ER stress sensitivity of mouse embryonic fibroblasts (MEFs), deficient for both iRhom1 and iRhom2 (iRhom1$^{-/-}$; iRhom2$^{-/-}$ double knockout, from here referred to as iRhom1/2 DKO)[11,26]. As expected, prolonged ER stress induced by treatment with tunicamycin (which inhibits protein glycosylation) or brefeldin A (which triggers collapse of the Golgi apparatus into the ER) caused wild-type cells to die, as measured by double staining with annexin V and propidium iodide. These cells showed typical apoptotic morphology, including a reduced cell volume and fragmented nucleus. Cell death under these conditions was strongly suppressed in iRhom1/2 DKO MEFs (Fig. 1a). Similar results were obtained using a second cell death assay, measuring cytotoxicity using lactate dehydrogenase (LDH) release into the cell culture medium (Fig. 1b). The iRhom1/2 DKO MEFs were not fully resistant to ER stress-induced apoptosis: by 36 h post-treatment, cell death was pronounced, albeit still at a significantly lower level than in wild-type cells (Supplementary Fig. 1a). Another canonical measure of cell death is the cleavage of poly (ADP-ribose) polymerase-1 (PARP-1) into several fragments[27]. Generation of the cleaved fragments of PARP by tunicamycin-induced ER stress was reduced in the absence of either iRhom1 or iRhom2, and loss of both iRhoms very strongly reduced PARP-1 cleavage (Fig. 1c). Deglycosylation of nicastrin acted as a positive control for tunicamycin (Tuni) treatment. This suggests a partially overlapping role for iRhom1 and iRhom2 in regulating cell death under ER stress conditions. Deletion of both iRhoms from a different cell line, A549 human adenocarcinomic alveolar basal epithelial cells, showed a similar reduction in cleaved PARP-1 in the absence of iRhoms (Supplementary Fig. 1b). Finally, deletion of iRhom2 in BV2a mouse microglia cells, which have endogenously low expression of iRhom1, also showed a significant reduction in PARP cleavage upon tunicamycin exposure (Supplementary Fig. 1c). Together these results demonstrate that iRhoms participate in ER stress-induced cell death across multiple mammalian cell types.

We next turned to a physiological whole organism system, taking advantage of the viability of *Drosophila* mutants in the

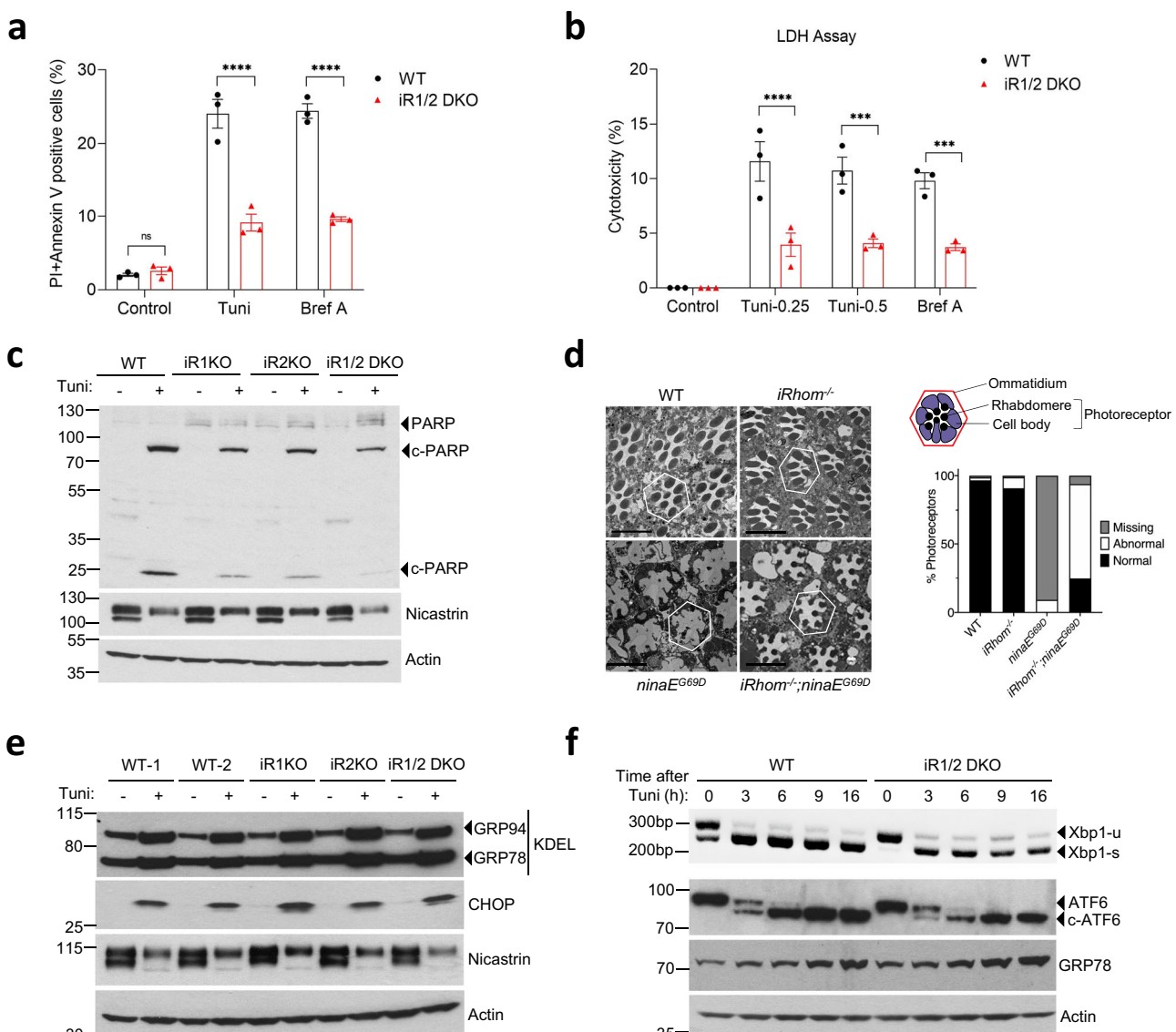

**Fig. 1 iRhoms regulate ER stress-induced cell death. a** Cell death of WT and iRhom1/2 DKO MEFs was measured after 18 h treatment with either tunicamycin (Tuni) (0.5 μg/ml) or brefeldin A (Bref A) (1 μg/ml) by PI/Annexin V staining using flow cytometry. Percentages of dead cells positive for both PI and Annexin V are shown in bar chart (n = 3, biologically independent experiments). Data are expressed as mean ± SEM. Two-way ANOVA (Sidak's), ****p < 0.0001 and ns = not significant. **b** Cell death of WT and iRhom1/2 DKO MEFs was determined after 18 h treatment with either tunicamycin (0.25 μg/ml or 0.5 μg/ml) or brefeldin A (1 μg/ml) by lactate dehydrogenase (LDH) assay. Lactate dehydrogenase release in the extracellular medium was quantified and percentage cytotoxicity was determined and shown in bar chart (n = 3, biologically independent experiments). Data are expressed as mean ± SEM. Two-way ANOVA (Sidak's), ***p < 0.001 and ****p < 0.0001. **c** Cell lysates of WT, iRhom1 and iRhom2 knockout MEFs after exposure to tunicamycin (0.5 μg/ml, 18 h) were analysed by immunoblotting with indicated antibodies. Black triangles denote full-length and cleaved proteins. **d** Representative transmission electron microscopy (TEM) images of 4-week old adult *Drosophila* retinas showing the overall structure of groups of ommatidia in a WT (n = 34), *iRhom*[−/−] (n = 204), *ninaE*[G69D] (n = 33) and *iRhom*[−/−]; *ninaE*[G69D] (n = 105) mutant flies, from two biologically independent experiments. Bar chart shows quantification of degeneration measured as the percentage of normal, abnormal and missing photoreceptors in each genotype indicated. Schematic of an ommatidium highlights its cellular structure. Scale bar = 10 μM. **e** Cell lysates of WT, iRhom1 and iRhom2 knockout MEFs after exposure to tunicamycin (0.5 μg/ml, 18 h) were analysed by immunoblotting with indicated antibodies. Anti-KDEL antibody detects both GRP94 and GRP78. **f** Splicing of *Xbp1-u* to *Xbp1-s* was determined using RT-PCR from mRNA (top panel) and cell lysates of WT and iRhom1/2 DKO MEFs exposed to tunicamycin (0.5 μg/ml) for the indicated times were analysed by immunoblotting with indicated antibodies (lower panels). Immunoblotting and RT-PCR data are representative of 2–3 biologically independent experiments. Source data are provided as a Source data file.

single *iRhom* gene[9]. Instead of chemical induction of ER stress, we used a well characterised genetic model, induced by a mutation in the *ninaE* gene (*ninaE*[G69D]), which encodes rhodopsin 1[28]. This mutant rhodopsin cannot fold properly, leading to ER stress in the photoreceptors, which in turn triggers cell death and retinal degeneration[29,30]. As expected, 4-week old *ninaE*[G69D] flies showed loss of photoreceptors and massive retinal

degeneration (Fig. 1d). This phenotype was significantly rescued in flies with a null *iRhom* mutation (the *iRhom* mutant flies have no significant eye defects). *ninaE*[G69D] flies lost about 90% of their photoreceptors, and the few remaining have abnormal morphology, whereas *iRhom*[−/−]; *ninaE*[G69D] flies retain 93% of their photoreceptors and 25% of them have normal morphology (Fig. 1d). This genetic result in flies supports the mammalian cell

culture experiments and confirms a physiological role for iRhoms in regulating ER stress-induced cell death.

The initial reaction of cells exposed to ER stress is to trigger the UPR signalling pathways[15,17]. We, therefore, investigated whether any of the three branches of the UPR, triggered by PERK, IRE1, and ATF6, respectively, were affected by the absence of iRhoms. Bearing in mind the significant interplay between the branches, we assessed outputs of each branch upon tunicamycin treatment: (i) upregulation of CHOP (primarily downstream of PERK) (Fig. 1e), (ii) splicing of $Xbp1$ (downstream of IRE1) (Fig. 1f, top panel), (iii) cleavage of ATF6 (Fig. 1f, lower panel), and (iv) upregulation of ER chaperones GRP94 and GRP78 (Fig. 1e, f). In each of these assays, the time- (Fig. 1f) and concentration-dependent (Supplementary Fig. 1d) effects of tunicamycin were normal in iRhom1/2 DKO MEFs. Notably, the level of CHOP, an ER stress-induced apoptotic transcription factor that is a major trigger of apoptosis[18–21], was unaffected by loss of iRhoms (Fig. 1e and Supplementary Fig. 1d). Consistent with this, upregulation of the CHOP target $Ero1l$[31] was also equivalent in wild-type and iRhom1/2 DKO MEFs (Supplementary Fig. 1e). These results demonstrate that, despite having an essential function in promoting ER stress-induced cell death in both flies and mammalian cells, iRhoms do not affect the UPR, indicating that iRhoms act either downstream or independently of the CHOP-induced death pathway.

**iRhoms trigger ER stress-mediated cell death via the intrinsic mitochondrial pathway**. Regardless of what initiates programmed cell death, apoptosis is primarily triggered by the caspases, which act via two distinct signalling pathways: the intrinsic (mitochondria-mediated) and the extrinsic (death receptor-mediated) pathways[32] (Fig. 2a). Both pathways converge on the activation of the effector caspases, including caspase-3, and cleavage of its substrates, including PARP. Tunicamycin-induced ER stress led to the cleavage of both caspase-3 and PARP in wild-type cells, but this cleavage was strongly suppressed in iRhom1/2 DKO MEFs (Fig. 2b). Strikingly, cleavage of the initiator caspase-9, an activator of caspase-3 was also inhibited (Fig. 2b). Caspase-9 activation is regulated by mitochondria (Fig. 2a), suggesting that iRhoms participate in ER stress-mediated cell death via the mitochondrial pathway. Reconstituting the iRhom1/2 DKO MEFs with both iRhom1 and iRhom2 restored caspase-3 and PARP cleavage to almost wild-type cells levels (Fig. 2c), confirming that the observed defect in apoptosis is specifically caused by the absence of iRhoms. iRhom1/2 DKO MEFs were also resistant to thapsigargin-evoked apoptosis, which, by inhibiting the SERCA $Ca^{2+}$ pump, leads to ER stress-induced apoptosis (Supplementary Fig. 2a). Non-ER stress stimuli can also induce $Ca^{2+}$-dependent apoptosis[33–35], and iRhom1/2 DKO MEFs were also less sensitive to etoposide, a DNA-damaging agent (Supplementary Fig. 2a). This result further implies that iRhoms are crucial in regulating the intrinsic mitochondrial apoptotic pathway.

A central element of the mitochondrial cell death pathway is a decrease in mitochondrial membrane potential followed by cytochrome c release[36]. Using the indicator dye TMRE, we observed a reduction in mitochondrial membrane potential in wild-type cells in response to tunicamycin or Brefeldin A; this was strongly suppressed in iRhom1/2 DKO MEFs (Fig. 2d). As a control, treatment with FCCP, a mitochondrial uncoupler, led to an extensive and equal reduction in mitochondrial membrane potential in both cell lines (Fig. 2d). iRhom1/2 DKO cells also showed reduced cytochrome c release into the cytoplasm compared to wild-type cells after tunicamycin treatment (Supplementary Fig. 2b), further supporting a role of iRhoms in ER stress-mediated cell death via the mitochondrial pathway.

The best characterised function of iRhoms to date are as regulatory cofactors of the metalloprotease ADAM17, and the release of one of its primary substrates, tumour necrosis factor (TNF), is defective in iRhom1/2 DKO cells[11,12,37,38]. TNF is also an important activator of caspases via the extrinsic death receptor pathway[39]. It has been shown that, as well as mitochondrial-dependent apoptosis, persistent ER stress can induce cell death through the extrinsic pathway via death receptor 5, a member of the family of TNF receptors[40]. We, therefore, investigated if ADAM17 and TNF defects could account for the cell death resistance of iRhom1/2 DKO MEFs in response to ER stress. ADAM protease inhibitors GI254023X (specific to ADAM10) and GW280264X (which inhibits both ADAM10 and ADAM17)[37] (Supplementary Fig. 2c), did not prevent cleavage of caspase-3 upon tunicamycin treatment of wild-type or iRhom1/2 DKO cells (Fig. 2e). This result confirms that ER stress-mediated apoptosis does not depend on ADAM17 activity. Moreover, direct activation of the death receptor signalling pathway, using a combination of TNF and cycloheximide, led to a comparable activation of initiator caspase-8, as well as apoptotic caspase-3 and caspase-9, in wild-type or iRhom1/2 DKO cells (Fig. 2f). This demonstrates that the TNF-mediated death receptor pathway is functional in the absence of iRhoms. Finally, the activation of caspase-9, which is also triggered by the extrinsic pathway via the effect of tBID on mitochondria (Fig. 2a)[32], was unaffected by loss of iRhoms. These results further confirm that there is no inherent defect in mitochondria-mediated caspase-9 activation via receptor-activated apoptotic stimuli in the absence of iRhoms.

Overall, these results demonstrate that the role of iRhoms in ER stress-induced apoptosis occurs specifically via the intrinsic mitochondrial pathway, and is independent of their regulation of ADAM17.

**Absence of iRhoms inhibits IP$_3$R-mediated ER-mitochondria $Ca^{2+}$ transfer**. As iRhoms are substantially localised in the ER, we investigated the link to the defect in mitochondrial membrane potential and subsequent cell death observed in iRhom1/2 DKO cells under ER stress. The ER is the main store of intracellular $Ca^{2+}$, and acute influx of ER $Ca^{2+}$ into the mitochondrial matrix leads to a decrease in mitochondrial membrane potential and apoptosis[41]. Accordingly, we observed that tunicamycin-induced cell death in wild-type cells was reduced in cells loaded with BAPTA, a $Ca^{2+}$ chelator, confirming that ER stress-induced cell death is dependent on intracellular $Ca^{2+}$ (Supplementary Fig. 3a).

IP$_3$Rs are $Ca^{2+}$ channels localised mainly in ER membranes, where they mediate $Ca^{2+}$ release evoked by extracellular stimuli in most animal cells[42,43]. To assess IP$_3$R activity, MEFs were stimulated with ATP, which evokes IP$_3$ production via P2Y purinergic receptors[44]. In wild-type cells, ATP-triggered a concentration-dependent increase in cytosolic [$Ca^{2+}$]; this was significantly reduced in iRhom1/2 DKO MEFs (Fig. 3a). Similarly, $Ca^{2+}$ release evoked by activation of B2 receptors by bradykinin (BK), which also mobilise $Ca^{2+}$ from intracellular stores through IP$_3$Rs[45], was also abolished in iRhom1/2 DKO MEFs (Fig. 3b).

A possible explanation for the loss of ER $Ca^{2+}$ release in iRhom1/2 DKO MEFs might be a reduction in ER $Ca^{2+}$ content. However, treatment of cells with thapsigargin (Fig. 3c) or the ionophore ionomycin (Supplementary Fig. 3b) in $Ca^{2+}$-free media showed that the ER $Ca^{2+}$ content was indistinguishable in wild-type and iRhom1/2 DKO MEFs. Furthermore, we confirmed that both cell lines were similarly loaded with the Calbryte-520 $Ca^{2+}$ indicator (Supplementary Fig. 3c). This implies that the defects in iRhom1/2 DKO MEFs are caused by reduced IP$_3$R-mediated $Ca^{2+}$ release.

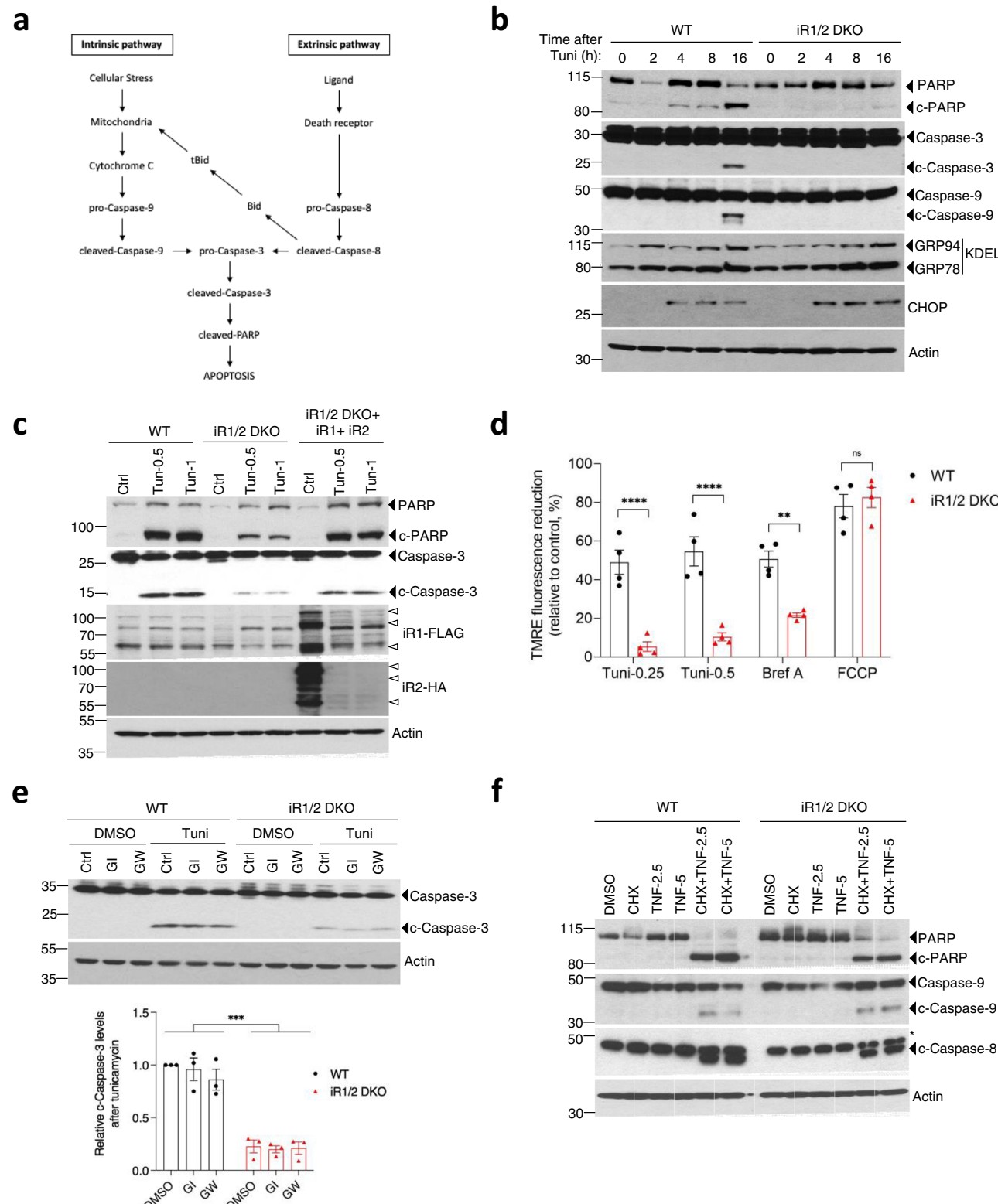

To examine IP₃R activity more directly, the intracellular stores of permeabilised MEFs were loaded with $Ca^{2+}$ and stimulated with IP₃. Consistent with previous results, iRhom1/2 DKO cells were less responsive than wild-type cells to IP₃ (Fig. 3d), although the difference was smaller than observed with GPCR-mediated stimulation (Fig. 3a, b) and was evident only with the highest concentrations of IP₃. We also used flash photolysis of caged-IP₃ to release IP₃ directly into intact cells. In these experiments,

iRhom1/2 DKO cells were slightly more responsive than wild-type cells to photolysis of caged-IP₃ (Fig. 3e). We return in the Discussion to this unexpected result, which contrasts with results from the same cells responding to IP₃ produced by endogenous signalling pathways stimulated by ATP or bradykinin. We note, however, that this result does not affect the conclusion that iRhoms regulate ER stress-induced cell death by inhibiting the $Ca^{2+}$-mediated mitochondrial apoptosis pathway.

**Fig. 2 iRhoms use the intrinsic apoptotic pathway under ER stress. a** Schematic showing the intrinsic and extrinsic cell death pathways regulated by caspases. **b** Cell lysates of WT and iRhom1/2 DKO MEFs after exposure to tunicamycin (0.5 μg/ml) for the indicated times were analysed by immunoblotting with indicated antibodies. **c** Cell lysates of WT, iRhom1/2 DKO and iRhom1/2 DKO MEFs stably reconstituted with iRhom1-3xFLAG and iRhom2-3xHA after exposure to tunicamycin (0.5 μg/ml or 1 μg/ml, 18 h) were analysed by immunoblotting with indicated antibodies. White triangle symbols denote major bands for iRhoms. **d** Mitochondrial membrane depolarisation in WT and iRhom1/2 DKO MEFs was measured using TMRE dye (250 nM) 18 h after treatment with tunicamycin (0.25 μg/ml or 0.5 μg/ml) or brefeldin A (1 μg/ml). FCCP (50 μM) was added for 15 min prior to measurement ($n = 4$, biologically independent experiments). Data are expressed as mean ± SEM. Two-way ANOVA (Sidak's), **$p < 0.01$ ****$p < 0.0001$ and ns = not significant. **e** Cell lysates of WT and iRhom1/2 DKO MEFs exposed to tunicamycin (0.5 μg/ml, 18 h) in the presence of ADAM protease inhibitors GI254023X (5 μM; specific to ADAM10) and GW280264X (5 μM; inhibits both ADAM10 and ADAM17) were analysed by immunoblotting with indicated antibodies. Bar chart shows quantification of cleaved caspase-3 relative to WT+DMSO as mean ± SEM ($n = 3$, biologically independent experiments). Two-way ANOVA (Sidak's), ***$p < 0.001$. **f** Cell lysates of WT and iRhom1/2 DKO MEFs treated for 18 h with cycloheximide (CHX 10 μg/ml) and/or TNF (2.5 ng/ml or 5 ng/ml) were analysed by immunoblotting with indicated antibodies. *Denotes unspecific band. Immunoblotting data are representative of 2–3 biologically independent experiments. Source data are provided as a Source data file.

---

We next investigated if ER Ca$^{2+}$ release triggered by tunicamycin is affected by the absence of iRhoms. ER stress stimulates IP$_3$Rs activity[46], and as expected, treatment of wild-type cells with tunicamycin in Ca$^{2+}$-free medium, leads to rapid and sustained oscillating release of ER Ca$^{2+}$ in wild-type cells (Fig. 3f). In contrast, there was no detectable ER Ca$^{2+}$ release in iRhom1/2 DKO MEFs. This marked inhibition of Ca$^{2+}$ release was comparable to the effect of treating wild-type cells with two inhibitors of IP$_3$Rs, 2-APB and Xestospongin C (Fig. 3f). While neither compound is an entirely selective inhibitor of IP$_3$Rs, their limitations are not shared[47]. These results nevertheless demonstrate a clear defect in Ca$^{2+}$ release by IP$_3$Rs in iRhom1/2 DKO cells under ER stress. Pre-treatment of cells with 2-APB also inhibited tunicamycin-induced death of wild-type cells to a level similar to that observed in iRhom1/2 DKO MEFs (Fig. 3g), further supporting the idea that loss of iRhoms has similar consequences to inhibition of IP$_3$Rs.

As well as releasing ER Ca$^{2+}$ into the cytoplasm, IP$_3$Rs can also deliver Ca$^{2+}$ directly to mitochondria at ER-mitochondrial membrane contact sites[41,48]. We, therefore, assessed if the observed defect in tunicamycin-induced IP$_3$Rs-mediated release of ER Ca$^{2+}$ had any effect on mitochondrial Ca$^{2+}$ influx. A genetically encoded fluorescent Ca$^{2+}$ indicator targeted to the mitochondria (CEPIA2mt)[49] was used to measure mitochondrial Ca$^{2+}$ in wild-type and iRhom1/2 DKO MEFs. There was a sharp increase in mitochondrial Ca$^{2+}$ concentration in wild-type cells within 10 min of tunicamycin treatment (Fig. 3h), which decreased gradually over time, but was maintained above the basal level for up to 2 h. In contrast, there was no rapid spike of mitochondrial Ca$^{2+}$ concentration in iRhom1/2 DKO MEFs, with only a small increase after 20 min, followed by a return to basal level by 1 h after treatment (Fig. 3h). These data demonstrate a requirement for iRhoms in the acute transfer of Ca$^{2+}$ from ER to mitochondria under conditions of ER stress.

Overall, these results lead us to conclude that loss of iRhoms causes diminished release of ER Ca$^{2+}$ through IP$_3$Rs, reduced mitochondrial Ca$^{2+}$ uptake, and thereby resistance ER stress-evoked cell death.

**IP$_3$Rs interact with iRhoms.** iRhoms are polytopic membrane proteins and all their known physiological functions rely on physical interaction with their clients, often other membrane proteins. We, therefore, examined whether iRhoms bind to IP$_3$Rs. Co-immunoprecipitation (co-IP) assays in cells expressing tagged iRhom2 and IP$_3$R1 showed that the proteins interact reciprocally (Fig. 4a). Similar observations were made using iRhom1 and IP$_3$R3 (Supplementary Fig. 4a). The specificity of this interaction was indicated by co-IP assays using iRhom1/2 DKO MEFs stably reconstituted with tagged iRhom2: iRhom2 was bound to endogenous IP$_3$R1 but not to other ER membrane Ca$^{2+}$ signalling

regulators (STIM1 and SERCA2) (Fig. 4b). We also used a proximity ligation assay (PLA) to examine association between iRhoms and IP$_3$Rs. This method detects molecules that are close enough to each other (about 40 nm) to allow a PCR reaction to occur between primer-tagged antibodies; the read-out is discrete fluorescent dots. A significantly higher signal was observed when using antibodies against endogenous IP$_3$R1 and N-terminally tagged iRhom2 expressed under a tetracycline-inducible promoter in iRhom1/2 DKO cells (Fig. 4c). An antibody against IP$_3$R3 gave a similar result (Supplementary Fig. 4b). These PLA results are consistent with the co-IP data, and together they support the conclusion that iRhoms physically interact with both tagged and endogenous IP$_3$Rs.

To characterise this interaction further, we tested IP$_3$R1 binding to iRhom2 mutants in which either the N-terminal cytoplasmic domain (iR2-ΔN) or the luminal iRhom homology domain (iR2-ΔIRHD) was deleted. Both of these forms bound to IP$_3$R1 indistinguishably from wild-type iRhom2 (Fig. 4d), indicating that these domains were dispensable for interaction. iRhoms interact with other membrane proteins primarily via their transmembrane domains (TMDs)[6,10,50,51] so we generated a minimal iRhom2 mutant containing only the cytoplasmic N-terminal domain, the first TMD and the luminal IRHD (i.e. lacking TMD2-TMD7). This truncated iRhom2 (iR2_TM-D1_IRHD) had a similar expression level and ER localisation as the wild-type protein (Supplementary Fig. 4c), and it had undiminished ability to bind and degrade epidermal growth factor (EGF) (Supplementary Fig. 4d), a known client of iRhoms[9]. Nevertheless, iR2_TMD1_IRHD had significantly reduced binding with IP$_3$R1 (Fig. 4e). This indicates that the intact iRhom2 TMD region is required for full interaction with IP$_3$R1, as is the case with some other membrane protein clients of iRhoms, including ADAM17 (Supplementary Fig. 4e). Given the role of iRhoms in IP$_3$R-mediated Ca$^{2+}$ release caused by ER stress, we asked whether the interaction between iRhoms and IP$_3$Rs is affected by tunicamycin treatment; indeed, this led to increased association between IP$_3$R1 and iRhom2 (Fig. 4f).

Taken together, these data demonstrate a physical interaction between IP$_3$Rs and the transmembrane domains of iRhoms, and show that this binding is enhanced by ER stress.

**BCL-2 is involved in the regulation of IP$_3$Rs by iRhoms.** How might iRhoms control the release of ER Ca$^{2+}$ in response to IP$_3$R stimulation? We examined the expression of all three IP$_3$R subtypes in iRhom1/2 DKO MEFs. Endogenous protein level of IP$_3$R1 was upregulated, and IP$_3$R3 was modestly downregulated in two independently derived iRhom1/2 DKO MEFs, with no detectable change in IP$_3$R2 (Fig. 5a, left panel). Quantitative RT-PCR showed that the increase in IP$_3$R1 protein in iRhom1/2 DKO cells correlated with more IP$_3$R1 (*Itpr1*) transcript, with no changes for IP$_3$R2 (*Itpr2*) and IP$_3$R3 (*Itpr3*) (Fig. 5a, right panel).

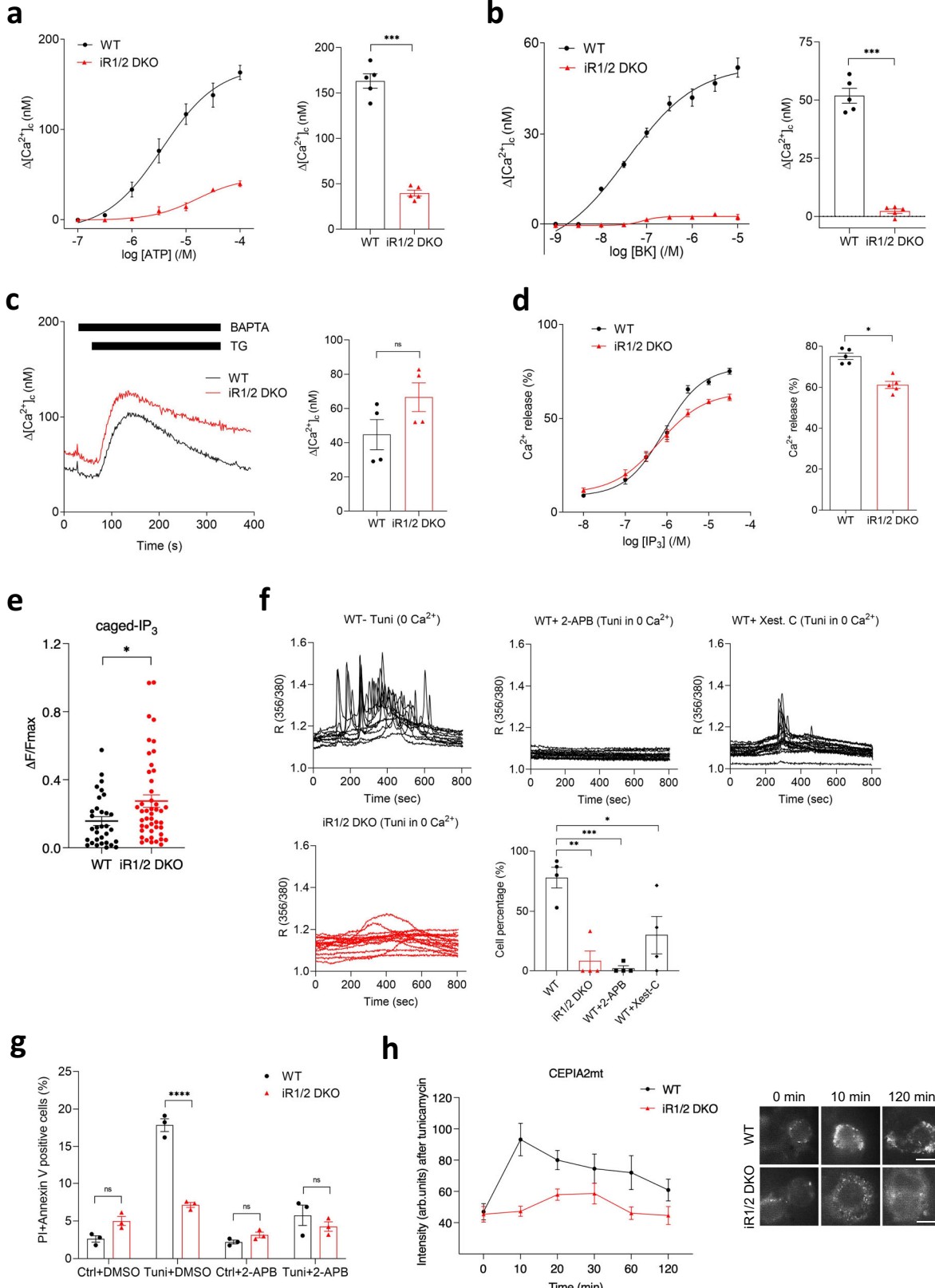

This increased IP$_3$R1 expression may be a compensatory response, but we have not investigated it further. Moreover, an increase in IP$_3$R1 in the absence of iRhoms could not easily explain the loss of ER Ca$^{2+}$ release. Conversely, the modest decrease in IP$_3$R3 protein could, in principle, contribute to the observed defect in iRhom1/2 DKO MEFs: IP$_3$R3 regulates ER Ca$^{2+}$ release and mitochondria-mediated apoptosis in some cell lines[52]. However, overexpression of IP$_3$R1 or IP$_3$R3 had no effect on cleaved caspase-3 formation in iRhom1/2 DKO MEFs treated with tunicamycin (Fig. 5b), implying that the lower level of IP$_3$R3 in iRhom1/2 DKO MEFs is not rate limiting, and is probably an indirect consequence of the loss of iRhoms.

**Fig. 3 iRhoms modulate ER stress-induced Ca$^{2+}$ release via IP$_3$Rs. a, b** IP$_3$-induced Ca$^{2+}$ release was measured in Calbryte 520-loaded cell populations of WT and iRhom1/2 DKO MEFs stimulated with various concentrations of ATP (**a**) or bradykinin (BK) (**b**) in Ca$^{2+}$-free media. Bar charts show the peak increase in cytosolic free [Ca$^{2+}$] ($\Delta$[Ca$^{2+}$]$_c$) evoked by 100 μM ATP (**a**) or 10 μM BK (**b**) in WT and iRhom1/2 DKO MEFs, mean ± SEM, $n = 5$ biologically independent experiments. Two-tailed paired Student's $t$ test, *** denotes $p < 0.001$. **c** ER Ca$^{2+}$ content was measured in populations of Calbryte 520-loaded WT and iRhom1/2 DKO MEFs stimulated with thapsigargin (TG, 1 μM) to release Ca$^{2+}$ from intracellular stores. BAPTA (2.5 mM) was added before stimulation to chelate extracellular Ca$^{2+}$. Bar chart shows the peak increase in [Ca$^{2+}$]$_c$ evoked by peak thapsigargin ($\Delta$[Ca$^{2+}$]$_c$), mean ± SEM, $n = 4$ biologically independent experiments. Two-tailed unpaired Student's $t$ test, ns = not significant. **d** Concentration-dependent effects of IP$_3$ on Ca$^{2+}$ release from permeabilised WT and iRhom1/2 DKO cells recorded using Mag-fluo 4 within ER lumen. Bar chart shows peak Ca$^{2+}$ release (%) as mean ± SEM, $n = 5$ biologically independent experiments. Two-tailed paired Student's $t$ test, * denotes $p < 0.05$. **e** Graph shows analyses of peak amplitudes of the changes in Calbryte 520-loaded WT and iRhom1/2 DKO MEFs fluorescence ($\Delta F/F_{max}$) evoked by photolysis of ci-IP$_3$, mean ± SEM, WT ($n = 32$), iR1/2 DKO ($n = 46$) from $n = 3$ biologically independent experiments. Two-tailed unpaired Student's $t$ test, * denotes $p < 0.05$. **f** ER Ca$^{2+}$ release in WT and iRhom1/2 DKO MEFs was measured using Fura 2 in Ca$^{2+}$-free medium after treatment with tunicamycin (0.5 μg/ml) for indicated times. WT cells were pre-treated for 2 h with IP$_3$R inhibitors, Xestospongin C (3 μM) or 2-APB (100 μM). Each trace represents a single-cell measurement. Bar chart shows responsive cells (%), mean ± SEM, $n = 12$ (WT), $n = 15$ (iR1/2 DKO), $n = 15$ (WT+2-APB), $n = 19$ (WT+Xest-C) from two biologically independent experiments. Ca$^{2+}$ signals are presented as 356 nm/380 nm fluorescence ratio (R). One-way ANOVA (Tukey's) *$p < 0.05$, **$p < 0.01$ and ***$p < 0.001$. **g** Cell death of WT and iRhom1/2 DKO MEFs was determined after treatment with tunicamycin (0.5 μg/ml, 18 h) alone or with 2-APB (100 μM) by PI/ Annexin V staining using flow cytometry. Percentage of dead cells positive for both PI and Annexin V are shown in bar chart ($n = 3$, biologically independent experiments). Data are expressed as mean ± SEM. Two-way ANOVA (Sidak's), ****$p < 0.0001$ and ns = not significant. **h** Concentrations of Ca$^{2+}$ in mitochondria of WT and iRhom1/2 DKO MEFs were measured using a targeted Ca$^{2+}$ indicator (CEPIA2mt) after exposure to tunicamycin (0.5 μg/ ml) for the indicated times. Intensity was calculated in arbitrary units (arb. units) from images of cells expressing the indicator ($n = 5$ images at each time point, 2 cells per image, from two biologically independent experiments) using ImageJ. Data are expressed as mean ± SEM. Representative images showing expression of CEPIA2mt as fluorescent puncta at selected time points. Scale bar = 10 μm. Source data are provided as a Source data file.

If changes in IP$_3$R expression do not explain the effect of iRhoms in regulating ER stress-induced Ca$^{2+}$ release and cell death, we must look elsewhere. BCL-2 family proteins are important regulators of IP$_3$R function[53]. Several antiapoptotic members (BCL-2, BCL-XL and MCL-1) bind to, and regulate the activity of IP$_3$Rs[53]. Immunoblot analysis showed modestly increased protein levels of BCL-2 and BCL-XL in iRhom1/2 DKO MEFs relative to wild-type cells, but no effect on MCL-1 proteins (Fig. 5c, top and lower left panels). The expression of *bcl-2*, but not *bcl-xl*, transcript was also marginally increased (Fig. 5c, lower right panel); we suspect this reflects a compensatory mechanism in iRhom1/2 DKO MEFs. Tunicamycin treatment caused a decrease of MCL-1, but had no effect on BCL-2 and BCL-XL protein levels (Fig. 5c, top panel). We next investigated whether BCL-2 or BCL-XL might contribute to the apoptosis-resistant phenotype of iRhom1/2 DKO MEFs under ER stress. We found that siRNA-mediated knockdown of *Bcl-2* in iRhom1/2 DKO MEFs treated with tunicamycin substantially rescued cleavage of caspase-3, to almost the same level as wild-type controls (Fig. 5d). A similar experiment knocking down *Bcl-xl* did not rescue cleavage of caspase-3 in iRhom1/2 DKO MEFs (Supplementary Fig. 5a), pointing to a specific role for BCL-2.

The rescue of the iRhom1/2 DKO phenotype by knockdown of *Bcl-2* suggests that iRhoms may regulate the inhibition of IP$_3$Rs by BCL-2. We, therefore, looked more closely at the relationships between these three proteins. BCL-2 inhibits IP$_3$R by direct binding[54–57] and we have shown that iRhom2 interacts with IP$_3$R1 and IP$_3$R3 (Fig. 4; Supplementary Fig. 4a). Co-IP assays using iRhom1/2 DKO MEFs stably reconstituted with tagged iRhom2 showed that it can bind endogenous BCL-2, IP$_3$R1 and IP$_3$R3, but not BCL-XL (Fig. 5e). We then tested iRhom2 truncations in which the N-terminal 100, 200 or 300 amino acid residues of the iRhom2 cytoplasmic domain was removed; the location of these truncated forms in the ER was indistinguishable from wild-type iRhom2 (Supplementary Fig. 5b). Co-IP of iRhom2 with BCL-2 was normal in the Δ100 construct, but substantially reduced in the Δ200 and Δ300 constructs (Fig. 5f), implying that a major determinant of the binding of BCL-2 to iRhom2 lies between residues 100 and 200 of the cytoplasmic domain. Finally, BCL-2 protein is located in several cellular compartments, so we investigated which pool binds to iRhoms.

Co-IP assays using wild-type BCL-2, ER-targeted BCL-2 (the C-terminal targeting domain of BCL-2 was replaced with the ER-targeting domain of cytochrome $b_5$)[58], and mitochondria-targeted BCL-2 (the C-terminal targeting domain of BCL-2 was replaced with the mitochondrial-targeting domain of monoamine oxidase B)[58] showed greater iRhom2 binding to ER-localised BCL-2 relative to wild-type BCL-2 and, significantly, no binding to mitochondrial BCL-2 (Fig. 5g). This is consistent with our conclusion of a functional interaction in the ER between iRhoms and BCL-2.

Taken together, these data indicate that BCL-2 plays an important role in the regulation of IP$_3$Rs by iRhoms, and that the three proteins form a network of regulatory interactions.

**iRhoms regulate binding of inhibitory BCL-2 to IP$_3$Rs.** The inhibitory effect of BCL-2 on IP$_3$Rs is dependent on a specific interaction between the BH4 domain of BCL-2 and a segment of the regulatory domain of IP$_3$Rs[54,57,59]. A peptide derived from this IP$_3$R domain, named BIRD-2 (BCL-2/IP$_3$ receptor Disruptor-2), specifically disrupts the IP$_3$R/BCL-2 complex, thereby promoting pro-apoptotic Ca$^{2+}$ signalling[60–62] (Supplementary Fig. 6a). Treatment of iRhom1/2 DKO MEFs with BIRD-2 rescued tunicamycin-induced caspase-3 cleavage to similar levels seen in wild-type cells (Fig. 6a). This striking result demonstrates that under ER stress conditions, iRhoms are required to relieve the inhibition of IP$_3$Rs by BCL-2, thereby enabling Ca$^{2+}$-dependent mitochondria-mediated cell death.

Tunicamycin-induced cell death in iRhom1/2 DKO MEFs was also rescued with the selective BCL-2 inhibitor, ABT-199/ Venetoclax (Fig. 6a). ABT-199 is a BH3-mimetic and induces apoptosis in a Ca$^{2+}$-independent manner by displacing pro-apoptotic members of the BCL-2 family from the hydrophobic pocket of anti-apoptotic BCL-2, to promote mitochondrial pore formation and release of cytochrome c[63]. This result shows that the defective ability of IP$_3$Rs to trigger pro-apoptotic Ca$^{2+}$ release in the absence of iRhoms can be rescued by directly activating the mitochondrial death pathway downstream of IP$_3$Rs.

As we have shown that iRhoms, BCL-2, and IP$_3$Rs form a regulatory network, we asked whether disrupting IP$_3$R/BCL-2 binding with BIRD-2 affected their interaction with iRhoms. Treatment with two concentrations of BIRD-2 peptide did not

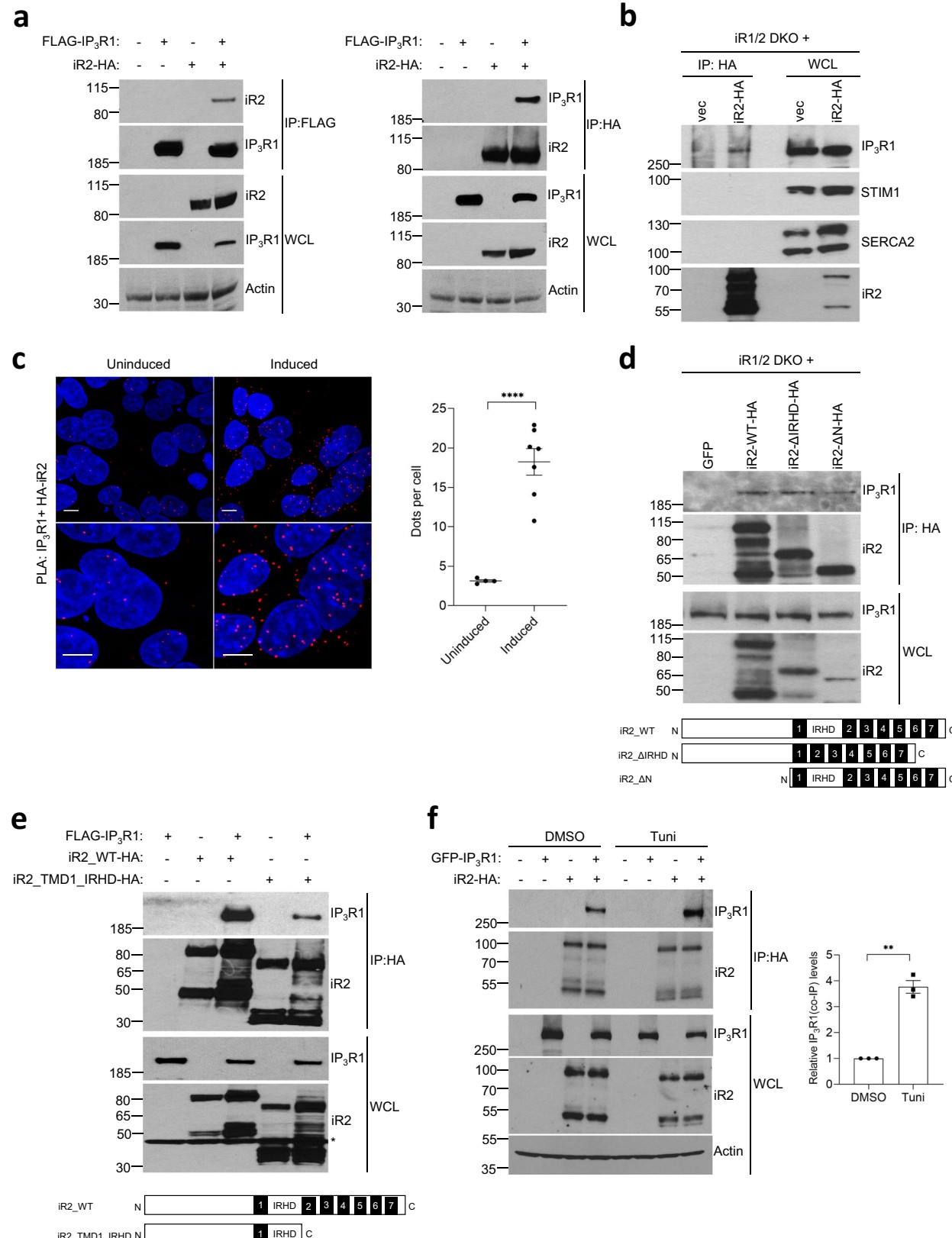

affect the binding of iRhom2 with endogenous BCL-2 (Fig. 6b), and nor did tunicamycin treatment affect the binding of these two proteins (Supplementary Fig. 6c). However, BIRD-2 increased the interaction between iRhom2 and endogenous IP$_3$R1 and IP$_3$R3 (Fig. 6b), as well as overexpressed IP$_3$R1 (Supplementary Fig. 6b). These results mirror those observed with tunicamycin (Fig. 4f).

Conversely, ABT-199 had no effect on the binding of iRhom2 with overexpressed IP$_3$R1 (Supplementary Fig. 6b), confirming that modulation of the interaction of iRhom2 with IP$_3$Rs is specifically caused by disrupting the BCL-2/IP$_3$R association. Finally, we assessed if iRhoms can regulate the interaction between IP$_3$Rs and BCL-2. In pull-downs of IP$_3$R1 in cells

**Fig. 4 iRhoms bind to IP$_3$Rs. a** Levels of iRhom2 and IP$_3$R1 were analysed by immunoblotting in whole cell lysate (WCL) and immunoprecipitation (IP: HA or IP: FLAG) from HEK293T cells transiently transfected for 36 h with FLAG-IP$_3$R1 and iRhom2-HA. **b** Levels of iRhom2 and endogenous IP$_3$R1, STIM1 and SERCA2 were determined by immunoblotting in whole cell lysate (WCL) and after immunoprecipitation (IP: HA) from iRhom1/2 DKO MEFs stably expressing iRhom2-HA. **c** Proximity ligation assay (PLA) in iRhom1/2 DKO HEK293T cells reconstituted with HA-iRhom2 under a tetracycline-inducible promoter were stained using antibodies to detect endogenous IP$_3$R1 and N-terminally HA-tagged iRhom2 after 24 h of 250 ng/ml of doxycycline to induce iRhom2. Dots per cell were quantified using ImageJ and shown as mean ± SEM, uninduced ($n = 4$ images/166 cells), induced ($n = 7$ images/259 cells) from two biologically independent experiments. Two-tailed unpaired Student's $t$-test ****$p < 0.0001$. Bottom panels are magnified sections of above corresponding panels. Scale bars = 10 µm. **d** Levels of WT and deletion mutants of iRhom2 and endogenous IP$_3$R1 were determined by immunoblotting in whole cell lysate (WCL) and immunoprecipitation (IP: HA) from iRhom1/2 DKO MEFs stably expressing iRhom2-WT-HA or iRhom2-ΔIRHD-HA or iRhom2-ΔN-HA. Schematics show iRhom2 mutants with domains deleted (numbering refers to TMDs and IRHD refers to iRhom homology domain). **e** Levels of WT and deletion mutant of iRhom2 and IP$_3$R1 were determined by immunoblotting in whole cell lysate (WCL) and immunoprecipitation (IP: HA) from HEK293T cells transiently transfected with iRhom2_WT-HA or iRhom2_TMD1_IRHD-HA with FLAG-IP$_3$R1 for 36 h. Schematics show iRhom2 mutant with domains deleted. * denotes unspecific band. **f** Levels of iRhom2 and IP$_3$R1 were analysed by immunoblotting in whole cell lysate (WCL) and after immunoprecipitation (IP: HA) from HEK293T cells transiently transfected with GFP-IP$_3$R1 and iRhom2-HA for 36 h, with tunicamycin (2 µg/ml) added for last 18 h. Bar chart shows quantification of co-IP GFP-IP$_3$R1 levels relative to total GFP-IP3R1 and shown as mean ± SEM ($n = 3$, biologically independent experiments). Two-tailed paired Student's $t$ test, ** denotes $p < 0.01$. Immunoblotting data are representative of 2–3 independent experiments. Source data are provided as a Source data file.

expressing tagged forms of iRhom2, IP$_3$R1, and BCL-2, the presence of iRhom2 reduced binding of IP$_3$R1 to BCL-2 (Fig. 6c). Similarly, in the presence of iRhom2, pull-downs of BCL-2 showed a decreased interaction with IP$_3$R1 (Fig. 6d). Combined with the data above, we conclude that iRhom2 interferes with the inhibitory interaction between BCL-2 and IP$_3$Rs, providing a molecular insight into the role of iRhoms in regulating ER stress-induced cell death via control of IP$_3$R-mediated Ca$^{2+}$ release.

## Discussion

iRhoms, which are located in the ER and secretory pathway, are the best studied non-protease members of the rhomboid-like superfamily of membrane proteins[5,7]. Here we used mammalian and *Drosophila* models to reveal an unexpected role for both iRhom1 and iRhom2 in ER homoeostasis, showing that they are specifically required for the apoptotic response to persistent ER stress. Cells deficient for iRhoms are resistant to the death that is normally caused by prolonged ER stress. In mammalian cells, we have shown that iRhoms are required for stress-induced transfer of Ca$^{2+}$ from the ER to mitochondria, a process that activates the intrinsic cell death pathway. Our results reveal that association of iRhoms with IP$_3$Rs and their inhibitor BCL-2 contributes to this failure of ER Ca$^{2+}$ release. Strikingly, knockdown of BCL-2 rescued ER stress-mediated cell death in cells lacking iRhoms, implying that iRhoms relieve BCL-2 inhibition of IP$_3$Rs. Our evidence that BIRD-2, a peptide that specifically inhibits BCL-2 binding to IP$_3$Rs[60–62], also rescues tunicamycin-induced apoptosis in cells without iRhoms, further supports our hypothesis. Overall, this work demonstrates for the first time a role of iRhom1 and iRhom2 in apoptosis triggered by unresolved ER stress. We propose that iRhoms are required to release IP$_3$Rs from inhibition by BCL-2, allowing IP$_3$Rs to deliver Ca$^{2+}$ to mitochondria and to thereby trigger apoptosis (Fig. 6e).

Control of Ca$^{2+}$ release by iRhoms via IP$_3$Rs is distinct from their best-known role in regulating the shedding of cell surface proteins by ADAM17, which accounts for their essential role in inflammation and EGF family growth factor signalling[8,9,11,12,37,38]. The relationship with ADAM17 is thought also to underlie the role of iRhoms in metabolic homoeostasis and obesity[25,64,65]. But iRhoms are already known to be multifunctional[5,7]. Beyond their regulation of ADAM17, iRhom2 has also been reported to regulate the trafficking and stability of STING, an essential adaptor in the pathway by which cells respond to DNA virus infection, by a mechanism that is ADAM17 independent[10]. And in other guises, the single *Drosophila* iRhom is implicated in the degradation of the membrane-

tethered precursors of EGF-family growth factors[9], and mammalian iRhom1 has been reported to affect proteasome stability under ER stress conditions[24]. Significantly, the effect on proteasome stability was observed shortly after exposure to ER stress, which corresponds to the adaptive phase of the UPR, distinct from our report here of a much later function in cells that are unable to recover from ER stress and consequently destined to die. Together, these observations demonstrate an integral role for iRhoms in the cellular response to ER stress.

Apoptosis occurs in multiple physiological, developmental, and stress contexts, and its misregulation is responsible for a variety of major pathologies[32]. One of its triggers is the toxic accumulation of unfolded proteins in the ER and our results show that cells deficient for both iRhom1 and 2 are extremely resistant to this form of cell death. An important mediator of apoptosis by the UPR is the transcription factor CHOP, which among other targets, stimulates the expression of ERO1-α (ER oxidase 1 α) to hyper-oxidise the ER lumen[31]. This is believed to activate IP$_3$Rs, which then release Ca$^{2+}$ to mitochondria[66] to initiate the intrinsic cell death pathway. Overall, this connection between the UPR and IP$_3$R-triggered Ca$^{2+}$ release is responsible for physiological ER stress-induced apoptosis[46]. In the absence of iRhoms, CHOP is induced and functional, but unable to activate apoptosis. We assume that other ER stress-induced apoptotic pathways, not involving IP$_3$Rs, remain functional but, in the absence of the major trigger, are insufficient to induce cell death. Indeed, many of these alternative pro-apoptotic signals ultimately converge onto the mitochondrial death pathway[15,19], which is attenuated in the absence of iRhoms. We note that iRhom1/2 DKO cells were also resistant to the DNA-damaging agent etoposide. Mitochondria-mediated apoptosis by some non-ER stress agents has also been shown to be partially dependent on intracellular Ca$^{2+}$ regulation[33–35] so, although we have not pursued the idea, it will be interesting to explore in the future whether iRhoms also have a role in regulating cell death caused by other stresses.

IP$_3$Rs are membrane-spanning channels that release Ca$^{2+}$ to the cytosol and, via specialised membrane contact sites, to the surface of mitochondria, allowing mitochondrial Ca$^{2+}$ uptake to regulate cell survival and death[41,67]. The three major IP$_3$R isotypes, IP$_3$R1, IP$_3$R2 and IP$_3$R3, share common structures and together orchestrate the release of Ca$^{2+}$ from the ER in a wide range of cellular processes[42,43]. We find that both iRhom1 and iRhom2 bind to IP$_3$Rs. This interaction requires the core transmembrane module of iRhoms, suggesting that, as for other clients of iRhoms[10,50,51], this function of iRhoms depends on specific TMD interactions. We have not yet mapped the site of interaction

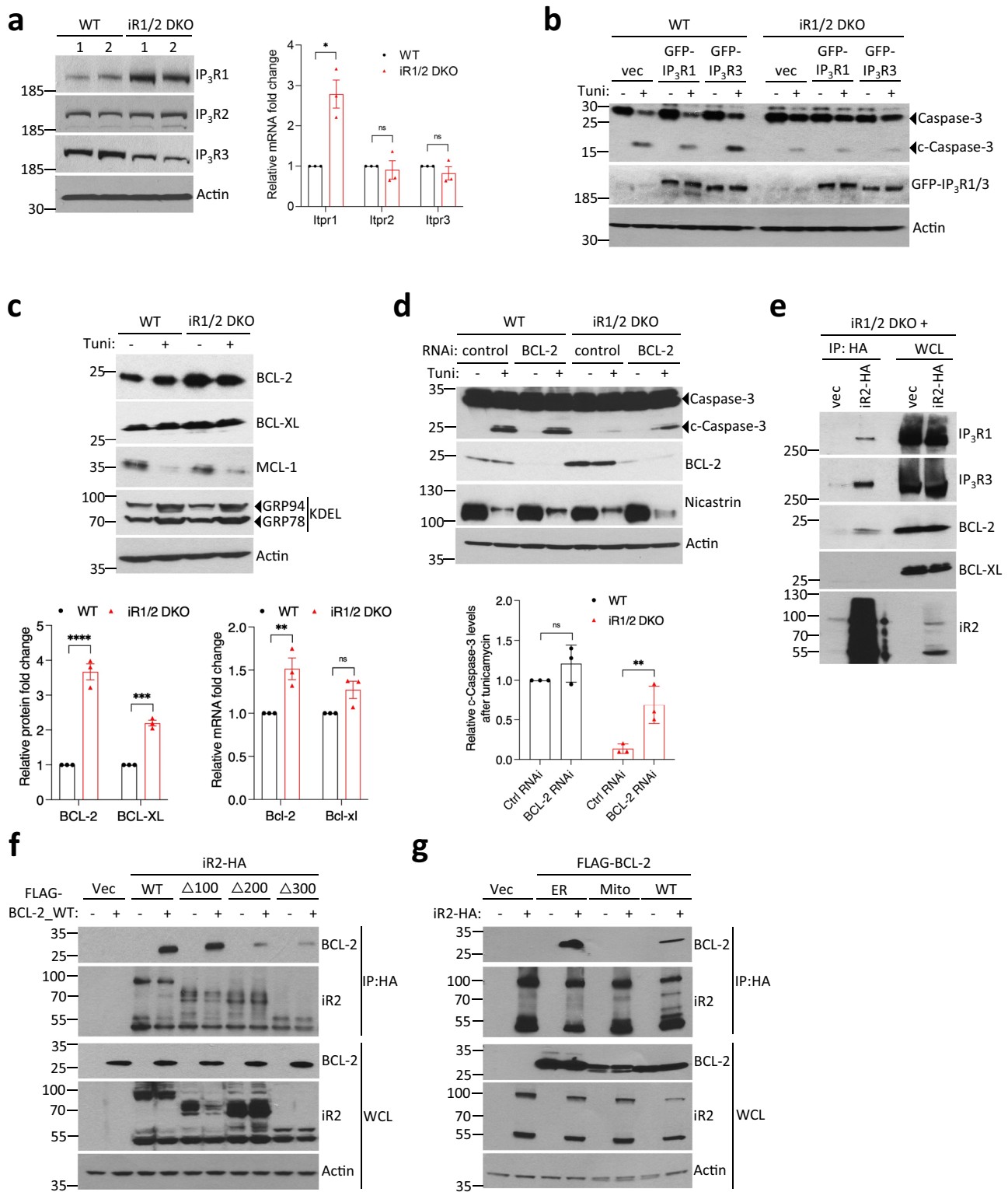

in the IP$_3$R: by analogy with other iRhom client proteins, we might expect this to be within one of its six TMDs, but this will need to be confirmed. Understanding the details of the interaction between iRhoms and IP$_3$Rs is motivated not only by a desire for full molecular understanding but also because of the possibility of future pharmacological modulation.

iRhoms also bind to BCL-2, in this case via the cytoplasmic N-terminal of iRhoms, specifically between amino acid residues 100 and 200. In contrast, the N-terminal region of iRhom2 is not required for binding to IP$_3$Rs, indicating that interaction of iRhom2 with BCL-2 and IP$_3$Rs may occur independently. BCL-2 is the founding member of the BCL-2 family of anti-apoptotic regulators, which are potential targets for cancer therapeutics[68]. As well as binding to and inhibiting pro-apoptotic members of the family (e.g. Bax and Bak), BCL-2 independently regulates the release of ER Ca$^{2+}$ [53,69] by binding to IP$_3$Rs. This promotes cell survival by two distinct mechanisms. First, BCL-2 is required for basal IP$_3$R-mediated transfer of Ca$^{2+}$ to mitochondria, which

**Fig. 5 BCL-2 forms a complex with iRhom2 and IP$_3$Rs to regulate ER stress-induced cell death. a** Cell lysates from two independent pairs of WT and iRhom1/2 DKO MEFs were analysed by immunoblotting with indicated antibodies (left panel). Levels of mRNA transcripts for all IP$_3$Rs in one pair of WT and iRhom1/2 DKO MEFs were determined by quantitative RT-PCR (right panel). Bar chart shows fold change relative to WT cells for each gene as mean ± SEM ($n = 3$, biologically independent experiments). Two-way ANOVA (Sidak's), * denotes $p < 0.05$, ns denotes not significant. **b** Cell lysates of WT and iRhom1/2 DKO MEFs transiently transfected for 36 h with GFP-IP$_3$R1 (2 μg) and GFP-IP$_3$R3 (2 μg), with tunicamycin (0.5 μg/ml) added for the last 18 h were analysed by immunoblotting with indicated antibodies. **c** Cell lysates from WT and iRhom1/2 DKO MEFs treated with tunicamycin (0.5 μg/ml, 18 h) were analysed by immunoblotting with indicated antibodies (top panel). Bar chart shows quantification of BCL-2 and BCL-XL proteins (bottom left panel) and mRNA transcripts levels determined by quantitative RT-PCR (bottom right panel). Data presented as fold change relative to WT cells as mean ± SEM ($n = 3$, biologically independent experiments). Two-way ANOVA (Sidak's), **** denotes $p < 0.0001$, *** denotes $p < 0.001$, ** denotes $p < 0.01$. **d** Cell lysates from WT and iRhom1/2 DKO MEFs transfected with control or *Bcl-2* RNAi for 72 h, with tunicamycin (0.5 μg/ml) added in the last 18 h were analysed by immunoblotting with indicated antibodies. Bar chart shows quantification of cleaved caspase-3 levels relative to its precursor after tunicamycin shown as mean ± SEM ($n = 3$, biologically independent experiments), Two-way ANOVA (Sidak's), **$p < 0.01$ and ns = not significant (bottom panel). **e** Levels of iRhom2 and endogenous IP$_3$R1, IP$_3$R3, BCL-2 and BCL-XL were determined by immunoblotting in whole cell lysate (WCL) and immunoprecipitation (IP: HA) from iRhom1/2 DKO MEFs stably expressing iRhom2-HA. **f** Levels of iRhom2 and BCL-2 were analysed by immunoblotting in whole cell lysate (WCL) and immunoprecipitation (IP: HA) from HEK293T cells transiently co-transfected for 36 h with FLAG-BCL-2_WT (0.25 μg) in combination with wild-type or indicated N-terminal iRhom2 deletions constructs (0.75 μg). **g** Levels of iRhom2, BCL-2 (WT, ER-targeted, Mitochondria-targeted) were analysed by immunoblotting in whole cell lysate (WCL) and immunoprecipitation (IP: HA) from HEK293T cells transiently transfected for 36 h with iRhom2-HA (0.75 μg) in combination with FLAG-BCL-2_WT (0.25 μg), FLAG-BCL-2_ER (0.25 μg), and FLAG-BCL-2_Mito (0.25 μg). Immunoblotting data are representative of 2–3 independent experiments. Source data are provided as a Source data file.

maintains cell proliferation and energy production[55,56,70]. Second, BCL-2 binding to IP$_3$Rs inhibits elevated, pro-apoptotic Ca$^{2+}$ transfer to mitochondria, thereby preventing mitochondrial depolarisation and cell death[54,55,57,71]. Control of these dual functions of BCL-2 depends on where it binds to IP$_3$Rs, the relative abundance of the two proteins, and the intensity of IP$_3$R stimulation[53,71].

BCL-2 has four Bcl-2 homology (BH) domains, of which BH4 is primarily associated with binding to IP$_3$Rs[54,57,59]. The IP$_3$R domains that interact with BCL-2 are more complex, with three distinct regions that bind to BCL-2, each contributing to IP$_3$R inhibition[53,62,71]. A major site of interaction is the IP$_3$R central domain. This led to development of the BIRD-2 peptide, which competes for binding to the BH4 domain of BCL-2[60-62] to disrupt BCL-2/IP$_3$R complexes, and thereby to relieve the inhibition of IP$_3$Rs by BCL-2. Our striking result that BIRD-2 rescues the iRhom1/2 DKO apoptotic phenotype in MEFs provides direct evidence that iRhoms interfere with the interaction between BCL-2 and IP$_3$Rs at their central domain. Our results also show that in the absence of iRhoms, even maximal stimulation with ATP or bradykinin fails to evoke Ca$^{2+}$ release, implying that a recently reported alternative inhibitory site, where IP$_3$ binding to IP$_3$Rs relieves BCL-2 inhibition[71], is probably irrelevant to iRhom function. Significantly, although we have not pursued it, the reduced Ca$^{2+}$ release in iRhom1/2 DKO MEFs suggests that iRhoms may have roles in Ca$^{2+}$ homoeostasis that extend beyond stress-induced apoptosis.

Our data demonstrate that the Ca$^{2+}$ signalling that triggers the intrinsic apoptotic pathway is controlled by interplay between IP$_3$Rs, BCL-2, and iRhoms. We found that iRhom2 reduced the binding of BCL-2 with IP$_3$R1, providing a potential explanation of how iRhoms promote IP$_3$R activity. Nevertheless, modulation of IP$_3$Rs activity is complex and we do not yet fully understand how iRhom2 induces the dissociation of BCL-2 from IP$_3$R1. As binding of BCL-2 and iRhoms to IP$_3$Rs occur at different region of the protein, this dissociation may not involve simple steric hindrance by iRhoms. Whether iRhoms recruit cofactors which directly regulate the binding of BCL-2 to IP$_3$Rs is an interesting prospect that needs further investigation. We also observed that inhibition of BCL-2 binding to IP$_3$Rs with BIRD-2 increased the interaction of iRhom2 with IP$_3$Rs, similar to the increased binding of iRhom2 and IP$_3$R1 after tunicamycin-induced ER stress. Together these data suggest a regulatory network of interactions that are modulated by ER stress.

Our hypothesis that iRhoms act directly on IP$_3$Rs is challenged by results using caged-IP$_3$. We used three independent methods to stimulate IP$_3$Rs: GPCR stimulation by agonists (ATP and bradykinin), in which the loss of iRhom2 massively reduced the Ca$^{2+}$ signals (Fig. 3a, b); addition of IP$_3$ to permeabilised cells, where loss of iRhom2 caused a more modest decrease in IP$_3$-evoked Ca$^{2+}$ release Fig. 3d); and photoactivation of caged-IP$_3$, where loss of iRhom2 caused no detectable inhibition of Ca$^{2+}$ signals (Fig. 3e). We have no simple explanation for this apparent disparity. It may be that in MEFs, GPCRs and uniform release of caged-IP$_3$ deliver IP$_3$ to different subsets of IP$_3$Rs that differ in their regulation by iRhoms. It is also possible that iRhoms, in addition to a direct effect on IP$_3$Rs, affect the signalling pathway by additional mechanisms, for example by regulating the pathway between GPCRs and IP$_3$Rs. IP$_3$Rs are also regulated by Kras-induced actin binding protein (KRAP), which 'licences' IP$_3$Rs to respond to IP$_3$;[72] perhaps iRhoms also contribute to this level of regulation. Despite this complexity, it is worth recapping the evidence that supports the conclusion that iRhoms regulate ER stress-induced apoptosis by modulating IP$_3$R function, particularly under ER stress. (1) In the absence of iRhoms, we observe reduced ER Ca$^{2+}$ release by GPCR activation by ATP and bradykinin) (Fig. 3a, b), and by IP$_3$ in permeabilised cells (Fig. 3d). (2) There is reduced IP$_3$R-dependent tunicamycin-induced ER Ca$^{2+}$ release (Fig. 3f) and reduced Ca$^{2+}$ uptake into mitochondria in the absence of iRhoms (Fig. 3h). (3) Tunicamycin-induced apoptosis is inhibited by the IP$_3$R inhibitor 2-APB (Fig. 3g). (4) iRhom2 binds directly to endogenous IP$_3$Rs and not to other ER-localised Ca$^{2+}$ channels (Fig. 4b). (5) ER stress increases binding of iRhom2 to IP$_3$R1 (Fig. 4f). (6) iRhom2 regulates the binding of inhibitory BCL-2 to IP$_3$R1 (Fig. 6c, d). Finally, (7) tunicamycin-induced apoptosis in iRhom1/2 DKO cells is rescued by removing the inhibition of BCL-2 on IP$_3$Rs, using BCL-2 RNAi (Fig. 5d) or BIRD-2 peptide (Fig. 6a).

Zooming out to the bigger picture, we report a previously unappreciated role for iRhoms in regulating Ca$^{2+}$ signalling and ER stress-induced cell death. Together with their ascribed roles in TNF signalling, EGFR signalling, and viral responses, iRhoms are emerging as multifunctional players in a variety of medically relevant signalling pathways. Although we do not yet understand in detail the mechanistic theme that links these roles, our understanding of the rhomboid-like superfamily leads us to believe that specific interaction with, and regulation of, membrane proteins may underlie their functions. A limitation of this

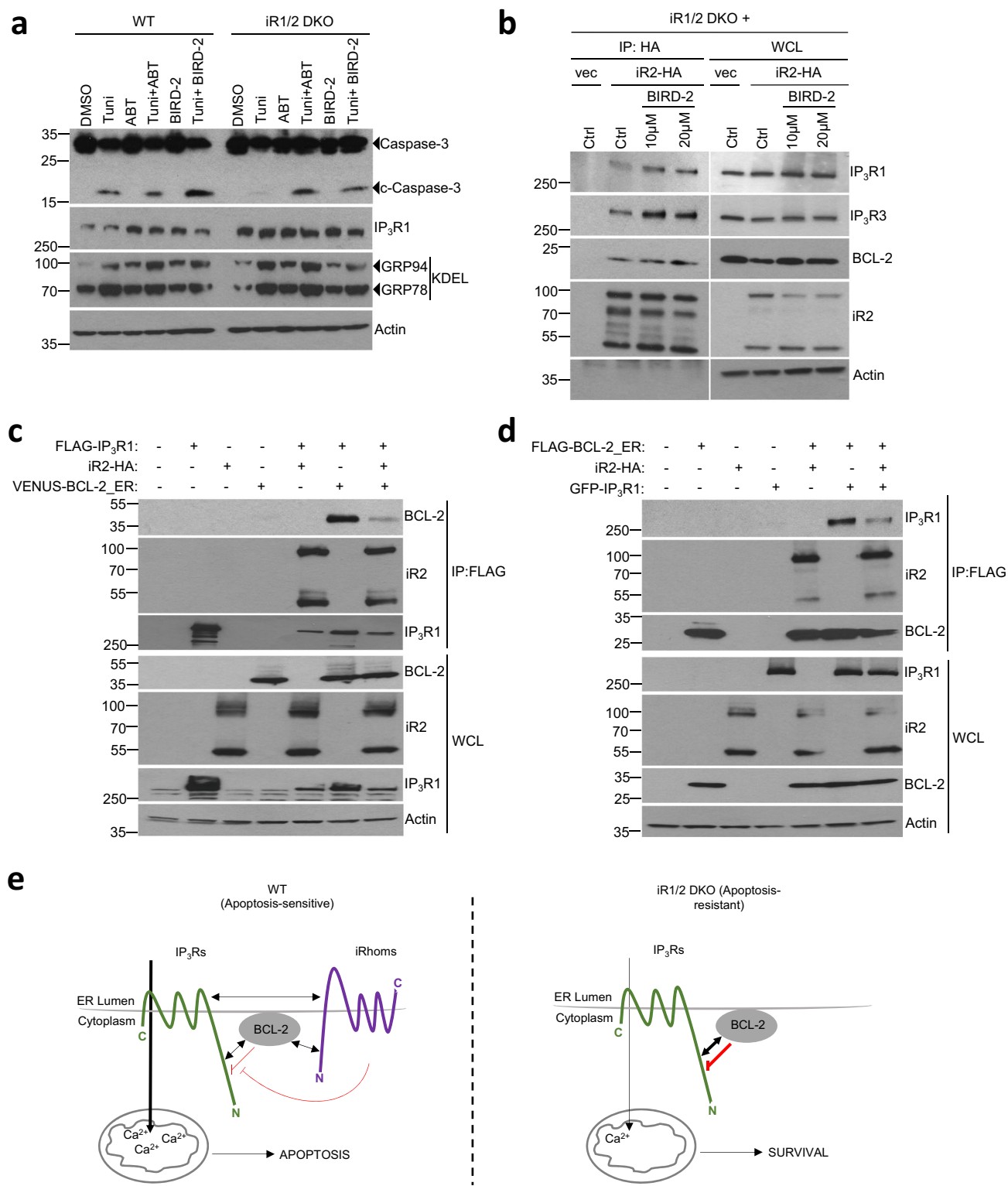

study has been our inability to examine the pathophysiological consequences of iRhom1/2 loss at the whole organism level in mammals, due to the embryonic lethality of iRhom1/2-deficient mice. We have partially addressed this by investigating genetic models of ER stress-induced cell death in *Drosophila*, and our results support the physiological importance of iRhoms in controlling cell death in response to ER stress. More directly, an obvious challenge for the future will be to understand the pathological significance of the role of iRhoms in Ca$^{2+}$ signalling

and cell death. For example, one interesting angle to pursue will be whether there is a relevance to Alzheimer's disease (AD), in which unresolved ER stress caused by misfolded proteins and deregulation of intracellular Ca$^{2+}$ signalling leads to neuronal cell death[73]. Intriguingly, recent reports show that *Rhbdf2* (coding for the iRhom2 protein) is among the top genes with differential level of CpG DNA methylation, particularly in early stages of AD[74–76], raising the possibility that the function of iRhoms we report here underlies this disease association.

**Fig. 6 iRhoms control BCL-2 antiapoptotic effect on IP$_3$Rs. a** Cell lysates from WT and iRhom1/2 DKO MEFs treated with tunicamycin (0.5 µg/ml) alone or in the presence of BIRD-2 (10 µM) or ABT-199/Venetoclax (ABT, 1 µM) were analysed by immunoblotting with indicated antibodies. **b** Levels of iRhom2 and endogenous IP$_3$R1, IP$_3$R3 and BCL-2 were analysed by immunoblotting in whole cell lysate (WCL) and immunoprecipitation (IP: HA) from iRhom1/2 DKO MEFs stably expressing iRhom2-HA after treatment for 18 h with control or BIRD-2 peptides (10 µM or 20 µM). **c, d** Levels of iRhom2, IP$_3$R1 and BCL-2 were analysed by immunoblotting in whole cell lysate (WCL) and immunoprecipitation (IP: FLAG) from HEK293T cells transiently transfected for 36 h with FLAG-IP$_3$R1 (1.25 µg), iRhom2-HA (0.25 µg) and VENUS-BCL-2_ER (0.25 µg; BCL-2 targeted specifically to ER) in (**c**) or GFP-IP$_3$R1 (1.25 µg), iRhom2-HA (0.25 µg) and FLAG-BCL-2_ER (0.25 µg; Bcl-2 targeted specifically to ER) in (**d**). Immunoblotting data are representative of 2–3 independent experiments. **e** Proposed model for the role of iRhoms in regulating IP$_3$R function via BCL-2. In WT cells, IP$_3$R, iRhoms and BCL-2 interact at the ER within a regulatory complex, where iRhom2 attenuates the inhibitory effect of BCL-2 on IP$_3$R activity. This enhances the flow of Ca$^{2+}$ from the ER into mitochondria under basal conditions and during acute release of Ca$^{2+}$ (e.g. ER stress) to regulate apoptosis. In the absence of iRhoms, the inhibitory effect of BCL-2 on IP$_3$Rs is maintained, thereby reducing ER-mitochondrial Ca$^{2+}$ transfer and enhancing cell survival. Source data are provided as a Source data file.

## Methods

**Molecular cloning.** iRhom1 was amplified from cDNA obtained from Mammalian Gene Collection by PCR and cloned with a C-terminal 3XFLAG tag into the retroviral vector pM6P.Histidinol using InFusion cloning according to manufacturer's protocol (Takara Bio, ♯639649). pM6P.Blasticidin plasmid expressing iRhom2 with a C-terminal 3XHA tag was previously described[38]. N-terminal truncated iR2 mutants (iR2-Δ100, iR2-Δ200, iR2-Δ300) and iRhom2_TMD1_IRHD were amplified by PCR from pM6P.Blast-miR2 and cloned with a C-terminal 3XHA tag into pEGFP-N1 mammalian expression vector (without EGFP protein). All primer sequences are provided in Supplementary Table 2.

**Cell culture and treatment.** Mouse embryonic fibroblasts (MEFs) from *Rhbdf1*$^{-/-}$, *Rhbdf2*$^{-/-}$ and *Rhbdf1*$^{-/-}$/*Rhbdf2*$^{-/-}$ (referred to by protein names iRhom1 KO, iRhom2 KO, and iRhom1/2 DKO, respectively) E13.5 embryos and wild-type C57BL/6J (RRID: IMSR_JAX:000664) controls, and immortalised by lentiviral transduction with SV40 large T antigen, have been previously described and isolated[11,26]. Ethical approval was obtained for these previous studies, but is not required for the subsequent use of the MEFs. iRhom1/2 DKO MEFs cells reconstituted with iR2_WT or iR2_ΔIRHD or iR2_ΔN have previously been described[38]. Human embryonic kidney (HEK293T) cells were obtained from ATCC (♯CRL-3216) and human adenocarcinomic alveolar basal epithelial cells (A549) were obtained from the Ervin Fodor lab, Oxford. A549- iRhom1/2 DKO and HEK293T-iRhom1/2 DKO cells were obtained from Boris Sieber. BV2a cells (ATCC, ♯CRL-2467) deleted of iRhom2 were generated using CRISPR/Cas9 system. Briefly, cells were nucleofected with Cas9 RNPs loaded with two guide RNAs with the following sequence: gRNA1-TAATACGACTCACTATAGGCGTTCCGTGCTAGATGCGA CGTTTTAGAGCTAGAAATAGCA and gRNA2-TAATACGACTCACTATAGG ACAGATATCCTCTTGCGGCGGTTTTAGAGCTAGAAATAGCA. Cells were selected with puromycin 48 h after transfection and single colonies were selected to establish clonal cell lines. Loss of iRhom2 was analysed by qRT-PCR.

HEK293T, MEFs, BV2a, and A549 cells were all cultured in high-glucose DMEM (Sigma-Aldrich, ♯D6429) supplemented with 10% foetal bovine serum (FBS) (Thermo Fisher Scientific, ♯10500064) and 5 mM glutamine (Gibco, ♯11539876) at 37 °C with 5% CO$_2$.

The following drugs were used: Tunicamycin (Santa Cruz, ♯sc-3506), Brefeldin A (Cell Signaling Technology, ♯9972), FCCP (Santa Cruz, ♯sc-203578), GI254023X (Sigma Aldrich, ♯SML0789), GW280264X (Aobious Inc, ♯AOB3632), Recombinant Mouse TNF-alpha (RayBiotech, ♯229-20286), cycloheximide (Sigma Aldrich, ♯C4859), Thapsigargin (Merck Chemicals, ♯586005), 2-APB (Tocris, ♯1224), Xestospongin C (AbCam, ♯ab120914), BAPTA-AM (Merck Chemicals, ♯CAS 126150-97-8), Fura2-AM (Thermo Fischer, ♯F1221), Calbryte™ 520-AM (AAT Bioquest, ♯20650), Ionomycin (Apollo Scientific, ♯BII0123), Doxycycline (Sigma Aldrich, ♯D9891), BIRD-2 (RKKRRQRRRGGNVYTEIKCNSLLPLAAIVRV) and control (RKKRRQRRRGGNVYTEGKCNSGGPLAAIGRV) peptides [Geert Bultynck and GenScript], ABT-199 (LKT Labs, ♯A0776), caged-IP$_3$ (ci-IP$_3$/PM: D-2,3-O-isopropylidene-6-O-(2-nitro-4,5-dimethoxy)benzyl-myo-inositol 1,4,5-trisphosphate hexakis (propionoxymethyl) ester) (SiChem, ♯cag-iso-2-145), D-myo-inositol 1,4,5-trisphosphate (IP$_3$) (Enzo Life Sciences, ♯ALX-307-007-M005). All drug concentrations are indicated in figure legends and in respective methods sections.

**Transfection and transduction of cell lines.** HEK293T cells and iRhom1/2 DKO MEFs were transiently transfected with DNA in OptiMEM (Gibco, ♯10149832) using FuGENE HD (Promega, ♯E2312) and Polyethylenimine (PEI) (Sigma Aldrich, ♯999012) respectively, and protein expression was analysed 24–48 h post transfection.

For knockdown experiments, siRNA was transfected using Lipofectamine RNAiMax (Thermo Fischer Scientific, ♯13778075) according to the manufacturer's protocol. ON-TARGETplus SMARTpool siRNA (Dharmacon) for mouse BCL-2 (L-063933-00-0005), mouse BCL-XL (L-065142-00-0005) and non-targeting siRNA control (Dharmacon; siGENOME D-001206-13-50) were used. Protein expression was analysed 72 h post siRNA transfection.

iRhom1/2 DKO MEFs cells stably expressing iRhom1-3XFLAG and iRhom2-3XHA were generated by retroviral transduction using pM6P retroviral constructs as previously described[38]. In brief, HEK293T cells were transfected with iRhom1 and iRhom2 expressed pM6P constructs together with packaging plasmid pCL-Eco. Viral supernatants for individual constructs were harvested after 48 h, cleared by centrifugation at 20,000 × *g* for 20 mins, and co-incubated with iRhom1/2 DKO MEFs cells in the presence of 5 µg/ml polybrene and cells were selected with blasticidin (Sigma Aldrich, ♯15205) and histidinol (Sigma Aldrich, ♯H6647). HEK293T-iRhom1/2 DKO + HA-iR2 TET-inducible cells were generated by lentiviral infection using N-terminally HA-tagged iRhom2 cloned into pLVX-TetOne vector (Clontech, ♯631849). Methodology is similar to retroviral transduction, with exception of packaging vectors (pCMV-VSV-G and pCMV-dR8.91) and selected with puromycin (Thermo Fischer Scientific, ♯A1113803).

**Antibodies.** For immunoblotting and co-immunoprecipitation: Actin (Santa Cruz, ♯sc-47778; 1:5000), ATF6 (Cell Signaling Technology, ♯65880; 1:1000), BCL-2 (AbCam, ♯ab182858; 1:500), BCL-XL (Cell Signaling Technology, ♯2764; 1:1000), Cleaved-Caspase-8 (Cell Signaling Technology, ♯8592; 1:1000), Caspase-3 (Cell Signaling Technology, ♯9665; 1:1000), Caspase-9 (Cell Signaling Technology, ♯9508; 1:1000), CHOP (Cell Signaling Technology, ♯2895; 1:250), GFP (AbCam, ♯ab13970; 1:2000), GRP78 (Cell Signaling Technology, ♯3177; 1:1000), FLAG-HRP (Sigma Aldrich, ♯A8592; 1:4000), HA-HRP (Roche, ♯11867423001; 1:2000), IP$_3$R1 (Thermo Fischer, ♯PA1-901; 1:1000), IP$_3$R1 & IP$_3$R2 [previously described[77]; 1:1000], IP$_3$R3 (BD Biosciences, ♯610312; 1:1000), KDEL (AbCam, ♯ab12223; 1:2000), MCL-1 (Cell Signaling Technology, ♯5453; 1:1000), Nicastrin (BD Biosciences, ♯612290; 1:1000), PARP (Cell Signaling Technology, ♯9542; 1:1000), SERCA2 (Cell Signaling Technology, ♯4388; 1:1000), STIM1 (Cell Signaling Technology, ♯5668; 1:1000).

For immunofluorescence: DAPI (Thermo Fischer, ♯D1306; 1 µg/ml), HA (Cell Signaling Technology, ♯3724; 1:500), BAP31 (Enzo Life Sciences, ♯ALX-804-601-C100; 1:250).

For proximity ligation assay: HA (Cell Signaling Technology, ♯3724; 1:500), HA (Enzo Life Sciences, ♯ENZ-ABS118-0200; 1:200), IP$_3$R1 (Thermo Fischer, ♯PA1-901; 1:200), IP$_3$R3 (BD Biosciences, ♯610312; 1:200).

**Plasmids.** Unless indicated, all iRhoms plasmids have previously been described[9,11,38]. FLAG-IP$_3$R1 construct was gifted from Marc Montminy, The Salk Institute for Biological Sciences. The EGFP-IP$_3$R1 and EGFP-IP$_3$R3 constructs were previously described[78]. VENUS-BCL-2_ER construct was gifted from David Andrews, University of Toronto. FLAG-BCL-2_WT, FLAG-BCL-2_ER and FLAG-BCL-2_Mito were obtained from Clark Distelhorst (Addgene plasmids ♯18003, ♯18004, ♯18005, respectively).

**SDS-PAGE and immunoblotting.** Cell lysates were denatured at 65 °C for 15 min and ran either on 4–12% NuPAGE™ Bis-Tris gels or 8–16% Tris-Glycine Novex™ WedgeWell™ (Thermo Fischer Scientific, ♯NP0321, ♯XP08160) in MOPS and Tris-Glycine running buffer respectively. PageRuler™ Plus Prestained Protein Ladder (Thermo Fischer Scientific, ♯26620) was used for protein molecular weight marker. Note this ladder runs differently on Bis-Tris and Tri-Glycine gels, resulting in different molecular weights according to manufacturer. Both type of gels were used throughout study and gels were transferred onto polyvinylidene difluoride (PVDF) membranes (Millipore, ♯IPVH85R). The membrane was blocked in 5% milk-TBST (150 mM NaCl, 10 mM Tris-HCl pH 7.5, 0.05% Tween 20, 5% dry milk powder) before incubation with indicated primary and species-specific HRP-coupled secondary antibodies. All primary antibodies were made in 5% BSA-TBST except for HRP-conjugated antibodies. Band visualisation was achieved with Amersham Enhanced Chemiluminescence (GE Healthcare, ♯RPN2106) or SuperSignal™ West Pico PLUS Chemiluminescent Substrate (Thermo Fisher Scientific, ♯34577) using X-ray film. Quantification of blots was done using Fiji (Image J).

**Co-immunoprecipitation.** HEK293T cells were transfected in 6-cm plates with indicated plasmids for 36–48 h before harvest. MEFs cells stably expressing iR2

were also processed according to the following steps. Cells were washed with ice-cold PBS and then lysed on ice in Triton X-100 lysis buffer (1% Triton X-100, 200 mM NaCl, 50 mM Tris-HCl pH 7.4) supplemented with cOmplete™, EDTA-free Protease Inhibitor Cocktail (Merck, ♯4693132001). Cell lysates were cleared by centrifugation at $21,000 \times g$ for 20 min at 4 °C. Protein concentrations were measured using Pierce™ Coomassie (Bradford) Protein Assay Kit (Thermo Fisher Scientific, ♯23236). Lysates were immunoprecipitated with 15 µl pre-washed Pierce™ Anti-HA Magnetic beads (Thermo Fisher Scientific, ♯88837) or Anti-FLAG® M2 Magnetic Beads (Sigma Aldrich, ♯M8823) at 4 °C overnight on a rotor. Beads were washed 4–5 times with Triton X-100 wash buffer (1% Triton X-100, 500 mM NaCl, 50 mM Tris-HCl pH 7.4) and proteins were eluted by incubation at 65 °C for 15 min in 2× SDS sample buffer.

For concanavalin A pull-down, N-glycosylated proteins were enriched by incubating cell lysates containing protease inhibitor and 1,10-phenanthroline (Sigma-Aldrich, ♯131377) with 20 µl concanavalin A Sepharose beads (Sigma-Aldrich, ♯ C9017) at 4 °C for at least 2 h with rotation. Beads were washed with Triton X-100 wash buffer and proteins were eluted in 2× NuPAGE™ LDS sample buffer (Thermo Fischer Scientific, ♯NP008) supplemented with 50 mM DTT and 50% sucrose for 15 min at 65 °C and were ran on 4–12 % NuPAGE™ Bis-Tris gels.

**Alkaline phosphatase shedding assay.** $5 \times 10^4$ MEF cells were seeded in a 24-well plate in triplicates 24 h before transfection. 500 ng alkaline phosphatase (AP)-conjugated amphiregulin (AREG) were transfected with Lipofectamine™ LTX Reagent with PLUS™ Reagent (Thermo Fisher Scientific, ♯15338100) according to the manufacturer's instructions. 24 h after transfection, cells were washed once with PBS and incubated for 18 h in 300 µl Phenol Red-free OptiMEM (Gibco, ♯11058-021) supplemented with 5 µM GW280264X (GW) or 5 µM GI254023X (GI) as indicated. The supernatants were then collected, and cells were lysed in 300 µl Triton X-100 lysis buffer supplemented with cOmplete™, EDTA-free Protease Inhibitor Cocktail (Merck, ♯4693132001). 100 µl supernatant and 100 µl diluted cell lysates were independently incubated with 100 µl AP substrate p-nitrophenyl phosphate (PNPP) (Thermo Scientific, ♯37620) at room temperature and the absorbance was measured at 405 nm by a plate reader (SpectraMax M3, Molecular Devices). The percentage of substrate release was calculated by dividing the signal from the supernatant by the total signal (supernatant and cell lysate).

***Xbp1* RT-PCR.** RNA was isolated from cells using the RNeasy Mini kit (Qiagen, ♯74104) and *Xbp1* bands (both spliced and unspliced) from MEFs were amplified directly with primers (Fwd-5′GAAGAGAACCACAAACTCCA 3′; Rev-5′GGA-TATCAGACTCAGAATCT 3′) using OneStep RT-PCR Kit (Qiagen, ♯210210). Fragments were analysed by gel electrophoresis using 2.5% agarose gel in TAE buffer. 26 base pairs are excised out from unspliced *Xbp1* to yield the spliced fragment.

**Quantitative RT-PCR.** RNA was isolated from cells using the RNeasy Mini kit (Qiagen, ♯74104) and reverse transcribed using the SuperScript™ VILO™ cDNA synthesis kit (Thermo Fischer Scientific, ♯11754050). Resulting cDNA was used for quantitative PCR (qRT-PCR) using the TaqMan™ Gene Expression Master Mix (Applied Biosystems, ♯4369016) and the following mouse TaqMan probes (Thermo Fisher Scientific): *ActB* (Mm02619580_g1), *Bcl-2* (Mm00477631_m1), *Bcl-xl* (Mm00437783_m1), *Ero1l* (Mm00469296_m1), *Itpr1* (Mm00439907_m1), *Itpr2* (Mm00444937_m1), *Itpr3* (Mm01306070_m1) and *Rhbdf2* (Mm00553470_m1). For quantification, the relative quantity of samples was calculated according the comparative ΔCt method and normalized to *ActB*. Gene expression was compared to the corresponding wild-type control.

**Cell death measurement by PI/Annexin V.** WT and iRhom1/2 DKO MEFs were plated in 6-well plates and treated with tunicamycin (0.5 µg/ml) or brefeldin A (1 µg/ml) at indicated time points. Cell death based on double staining for propidium iodide and Annexin V staining was measured using Annexin V-FITC Kit (Miltenyi Biotec, ♯130-092-052) according to manufacturer's protocol using BD FACSCalibur (BD Biosciences). FlowJo software was used for data analysis.

**Lactate dehydrogenase (LDH) assay.** WT and iRhom1/2 DKO MEFs were plated in 6-well plates and treated for 18 h with tunicamycin (0.25 µg/ml, 0.5 µg/ml) or brefeldin A (1 µg/ml) in reduced serum growth media OptiMEM (Gibco, ♯10149832). LDH release in extracellular media was measured using LDH Cytotoxicity Detection Kit (Takara Bio, ♯MK401) and percentage cytotoxicity was calculated according to the formula provided in manufacturer's protocol.

**Flies and transmission electron microscopy (TEM).** *iRhom*$^{-/-}$ flies have previously been described[9]. Wild-type flies ($w^{1118}$) were obtained from Bloomington Stock Centre and *nina*$^{EG69D}$ flies were a kind gift from Bertrand Mollereau and Pedro Domingos. All flies were maintained at 29 °C.

TEM imaging of *Drosophila* retina was performed as described[79]. Briefly, *Drosophila* heads were cut in half and fixed in 2.5% Glutaraldehyde, 4% PFA and 0.1% tannic acid in 0.1 M PIPES pH 7.2 for 1 h at room temperature (RT) and overnight at 4 °C. Samples were washed with 0.1 M PIPES at RT. Samples were

then incubated in 2% osmium tetroxide and 1.5% potassium ferrocyanide in 0.1 M PIPES for 1 h at 4 °C and washed 3 times with MQ water. This was followed by tertiary fixation with 0.5% uranyl acetate (aq.) at 4 °C overnight. Samples were washed 3 times with MQ water, then sequentially dehydrated in 30–100% ethanol and in anhydrous acetone. Samples were then infiltrated at RT with sequential mix of acetone and Durcupan resin. Samples were then embedded in the fresh resin and cured at 60 °C for 48 h. Eyes were sectioned tangentially using a Leica UC7 ultramicrotome. Ultrathin (90 nm) sections were post-stained for 5 min with Reynold's lead citrate and washed 3 times with warm water. Grids were imaged on a FEI Tecnai 12 Transmission Electron Microscope (TEM) at 120 kV using a Gatan OneView camera.

Missing, abnormal or missing photoreceptor rhabdomeres were counted manually on TEM images from a total of 33 to 204 ommatidia per genotype using Fiji (Image J).

**Mitochondrial membrane potential.** WT and iRhom1/2 DKO MEFs were plated in 6-well plates and treated for 18 h with tunicamycin (0.25 µg/ml, 0.5 µg/ml) or brefeldin A (1 µg/ml). FCCP (50 µM) was added for only 15 min. Changes in mitochondrial membrane potential of cells after the indicated treatments were measured using Tetramethylrhodamine ethyl ester (TMRE) Mitochondrial Membrane Potential Assay Kit (AbCam, ♯ab113852) and analysed according to the manufacturer's protocol using flow cytometry. For each genotype, the decrease in fluorescence after each treatment was calculated as a percentage relative to untreated cell populations. FlowJo software was used for data analysis.

**Ca$^{2+}$ measurements.** For measurements of [Ca$^{2+}$]$_c$ in single cells (Fig. 3f), cells were loaded with 1 µM Fura 2-AM (Thermo Fischer Scientific, ♯F1221) in external solution (145 mM NaCl, 2.8 mM KCl, 2 mM CaCl$_2$, 2 mM MgCl$_2$, 10 mM D-glucose, 10 mM HEPES, pH 7.4) for 40 min in the dark at 37 °C, and then were washed and incubated in external solution for another 15 min for full de-esterification. Cells were incubated in Ca$^{2+}$-free solution (145 mM NaCl, 2.8 mM KCl, 2 mM MgCl$_2$, 10 mM D-glucose, 10 mM HEPES, 0.1 mM EGTA) immediately prior to imaging measurements. 0.5 µg/ml tunicamycin was added directly onto cells immediately prior to measurements. The fraction of responsive cells was determined from cells in which the fluorescence ratio (R356/380) increased by >0.1 within 10 min. All the data were analysed by using IGOR Pro software.

For mitochondria Ca$^{2+}$ uptake (Fig. 3h), cells were transiently transfected with the CEPIA2mt Ca$^{2+}$ probe[49] for 48 h and treated with 0.5 µg/ml tunicamycin for the indicated time before images were taken of cells excited at 480 nm with emission recorded at 525 nm. Intensity of signal was quantified using ImageJ and plotted as arbitrary units (arb.units).

For measurements of [Ca$^{2+}$]$_c$ in cell populations (Fig. 3a–d), cells grown in 96-well plates were loaded with Calbryte 520 AM (AAT Bioquest, ♯20650) by incubation for 1 h at 20 °C in HEPES-buffered saline (HBS, 100 µl) containing Calbryte 520 AM (2 µM), 0.02% Pluronic F-127 (Sigma-Aldrich, #P2443) and probenecid (2.5 mM). Cells were then washed and incubated in HBS containing probenecid (2.5 mM) for 1 h at 20 °C to allow de-esterification of Calbryte 520 AM. HBS had the following composition: 135 mM NaCl, 5.9 mM KCl, 1.2 mM MgCl$_2$, 1.5 mM CaCl$_2$, 11.5 mm D-glucose, 11.6 mM HEPES, pH 7.3. BAPTA (final concentration 2.5 mM) was added to the HBS immediately before stimulation to reduce the free [Ca$^{2+}$] of the HBS to <20 nM, before addition of ionomycin (5 µM), thapsigargin, ATP and Bradykinin (BK). Calbryte 520 fluorescence was recorded using a FlexStation III fluorescence plate-reader that allows automated fluid additions while recording fluorescence (Molecular Devices). Fluorescence was recorded at 1.44-sec intervals, with excitation at 485 nm and emission at 525 nm. Data were collected and analysed using SoftMax Pro (Molecular Devices). Maximal ($F_{max}$) and minimal ($F_{min}$) fluorescence values were determined from parallel wells after addition of Triton X-100 (0.1%) to lyze cells with either 10 mM CaCl$_2$ ($F_{max}$) or 10 mM BAPTA ($F_{min}$). The fluorescence values ($F$) were calibrated to [Ca$^{2+}$]$_c$ using a $K_D = 1200$ nM (Calbryte 520) from:

$$[Ca^{2+}]_c = K_D \times \frac{(F - F_{min})}{(F_{max} - F)} \qquad (1)$$

IP$_3$-evoked Ca$^{2+}$ release in populations of permeabilised MEFs (Fig. 3d) used saponin to permeabilise the plasma membrane and Mag-fluo 4 AM (Thermo Fischer Scientific, #M14206), a low-affinity Ca$^{2+}$ indicator, to record [Ca$^{2+}$] within the ER. MEF cells were incubated with 20 µM Mag-fluo 4-AM in HBS containing BSA (1 mg/ml) and Pluronic F-127 (0.02%, v/v) for 1 h at 20 °C. Cells were then resuspended in Ca$^{2+}$-free cytosol like medium (CLM: 140 mM KCl, 20 mM NaCl, 1 mM EGTA, 2 mM MgCl$_2$ and 20 mM PIPES, pH 7.0) and permeabilised with saponin (10 µg/ml, 3 min, 37 °C). Cells were recovered by centrifugation ($600 \times g$, 2 min), resuspended ($10^7$ cells/ml) in Mg$^{2+}$-free CLM, distributed into pre-coated poly-L-lysine, black half-area 96-well plates (45 µl/well), and cells were collected at the bottom of the plates by centrifugation ($300 \times g$, 2 min). Mag-fluo 4 fluorescence (excitation at 485 nm, emission at 525 nm) was recorded at 1.44-s intervals at 20 °C using a FlexStation III fluorescence plate-reader. MgATP (1.5 mM) was added to allow uptake of Ca$^{2+}$ into the ER, and when the ER steady-state loading was achieved (~2 min), IP$_3$ was added together with cyclopiazonic acid (CPA, 10 µM) (Sigma-Aldrich, #C1530) a SERCA inhibitor, to prevent further Ca$^{2+}$ uptake. IP$_3$-

evoked $Ca^{2+}$ release is reported as the fractional decrease in the ATP-dependent Mag-fluo 4 fluorescence.

**Photolysis of caged-IP$_3$**. WT and iRhom1/2 DKO MEFs cells were grown on fibronectin coated glass-bottomed dishes. Cells were washed twice with HBS and incubated with Calbryte 520 AM (2 μM, 20 °C) for 1 hr in HBS containing 0.02% Pluronic F-127 and probenecid (2.5 mM). Cells were then loaded with ci-IP$_3$/PM (1 μM, 20 °C, 1 h) (SiChem, #cag-iso-2-145), washed twice and incubated in HBS for a further 30 min before imaging in HBS at 20 °C using an inverted Olympus IX83 microscope equipped with a ×60 objective. Calbryte 520 fluorescence was recorded in widefield (488 nm excitation, 525/50 nm emission), with illumination for 25 ms every sec. A flash of ultraviolet (UV) light (150 ms, 395/20 nm, SPECTRA X-light engine, Lumencor) was delivered after 10 s to photolyse ci-IP$_3$. Cells were imaged, background-corrected and analysed using MetaMorph (Molecular Devices).

**Proximity ligation assay (PLA)**. HEK293T iRhom1/2 DKO cells expressing N-terminal HA-tagged iRhom2 under tetracycline-inducible promoter were plated on 13-mm glass coverslips in 12-well plates and were treated with 250 ng/ml of doxycycline for 24 h to induce iR2 prior to fixation. Cells were washed 2 times with PBS and fixed with 4% paraformaldehyde in PBS at room temperature for 20 min. Cells were then washed 3 times with PBS and permeabilised in 0.3% TX-100 in PBS for 20 min. Coverslips were processed using Duolink® PLA Fluorescence kit (Sigma Aldrich, #DUO921101) according to manufacturer's protocol. Images were acquired with a laser scanning confocal microscope (Fluoview FV1000; Olympus) with a 60 × 1.4 NA oil objective and processed using Fiji (Image J).

**Immunofluorescence and confocal microscopy**. HEK293T cells were plated on 13-mm glass coverslips in 12-well plates and transfected with 100–250 ng of indicated constructs for 48 h prior to fixation. Cells were washed 2 times with PBS and fixed with 4% paraformaldehyde in PBS at room temperature for 20 min. Cells were then washed 3 times with PBS and permeabilised in 0.3% TX-100 in PBS for 20 min. Cells were blocked with 3% BSA in 50 mM Tris pH 7.5 for 30 min after removal of permeabilisation buffer. Cells were incubated overnight with indicated antibodies in blocking buffer at 4 °C and then washed 3 times in permeabilisation buffer for (5 min each wash). Coverslips were incubated with corresponding species-specific fluorescently coupled secondary antibodies (Invitrogen) for 30 min. Cells were subsequently washed 4 times with PBS (5 μg/ml DAPI was added in second to last wash), prior to mounting on glass slides with VECTASHIELD® anti-fade mounting medium (Vectorlabs, #H-1000-10). Images were acquired with a laser scanning confocal microscope (Fluoview FV1000; Olympus) with a 60×1.4 NA oil objective and processed using Fiji (Image J).

**Statistical analysis and data presentation**. Unless indicated, all data are expressed as mean ± SEM. Two-tailed Student's $t$ test, One-way ANOVA (Tukey's) and Two-way ANOVA (Sidak's) were used to compare differences between groups where indicated in the corresponding figure legends. GraphPad Prism 6 was used for statistical analysis, and statistical significance was set at a $p$-value of <0.05. In all the graphs, *, **, ***, ****, ns denotes $p$-value < 0.05, 0.01, 0.001, 0.0001, and not significant, respectively.

**Reporting summary**. Further information on research design is available in the Nature Research Reporting Summary linked to this article.

## Data availability
The data supporting the findings from this study are available within the article files and its Supplementary information. Source data are provided with this paper.

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

## Acknowledgements

We acknowledge members of the Freeman lab for their support and advice during the study. We are grateful to Kanaga Sabapathy for critical reading of the manuscript. We thank Vera Konieczny for help with exploring IP$_3$ receptor function early in the study. We also thank Marc Montminy (FLAG-IP$_3$R1 construct), David Andrews (VENUS-BCL-2_ER construct), Geert Bultynck (initial BIRD-2 and control peptides test samples), and Boris Sieber (HEK293T-iRhom1/2 DKO and A549-iRhom1/2 DKO cells) for generously sharing their reagents. This paper was supported by Wellcome Trust Senior Investigator Awards to M.F. (101035/Z/13/Z and 220887/Z/20/Z) and C.W.T (101844/Z/13/Z). P.A-A. is supported by a research fellowship from Emmanuel College, Cambridge.

## Author contributions

I.D. designed, performed and analysed most of the study and wrote the paper. P.A-A. performed and analysed Ca$^{2+}$ data (cell population measurements). Y.Y. performed and analysed Ca$^{2+}$ data (mitochondria and single cells). C.L and S.M performed and analysed fly and immunofluorescence data. F.L performed and analysed ADAM17 shedding assay. C.W.T guided and analysed the Ca$^{2+}$ data (cell population measurements). M.F. analysed and supervised the study, and wrote the paper with I.D. All authors have reviewed the paper.

## Competing interests

The authors declare no competing interests.

**Additional information**

