## [Peer Review File · Nature Communications]

REVIEWER COMMENTS

Reviewer #1 (Remarks to the Author):

iRhoms have emerged as key regulators of ADAM17-dependent TNF activation and their role in controlling ADAM17-mediated EGFR-signaling is well-established. However, although iRhoms have recently been associated with other biological functions, their exact roles in this regard have only recently begun to be elucidated in a limited number of studies.

In the current study, Dulloo et al. present compelling biochemical and cell biological evidence for the involvement of two catalytically inactive members of rhomboid proteases in ER stress-induced cell death through IP3 receptors and BCL-2 signaling pathways. The authors show that both iRhoms are necessary to regulate ER stress-induced calcium influx into mitochondria and that loss of iRhoms causes cells to be resistant to ER stress-induced apoptosis. They convincingly demonstrate that iRhoms are needed for the activation of apoptotic caspases, specifically via the intrinsic mitochondrial cell death pathway.

The authors deploy a wide range of elegant techniques to assess the role of these two inactive rhomboids in controlling apoptosis due to persistent ER stress.

The manuscript is well written and the observation that iRhoms control ER stress-induced cell death is novel.

I only have a few comments that the authors should address in their revised manuscript.

Introduction:

Line 63: Previous studies have provided evidence for the location of iRhoms in the endoplasmic reticulum. The authors should specify in their introduction whether these observations were based on overexpression experiments or detection of endogenous protein in mammalian cells and/or drosophila.

Results:

While the quality of the majority of western blots is excellent, especially when considering the challenges that are associated with Co-IPs, the data presentation in figure panel 2c is less convincing. Similarly to figure panel 2B, the “full-length” PARP should be shown as well. Also, the differences in cleaved caspase 3 levels are difficult to estimate in this particular blot and could potentially be replaced with another representative image. Alternatively, quantifications for figure panels with whole cell lysates could be provided.

Several structure/function studies have explored the role of individual segments of these multi-membrane spanning proteins. The authors’ observations that a truncated mutant containing only the cytoplasmic N-terminal domain, the first transmembrane domain as well as the luminal domain showed significantly reduced binding with IP3R1 is interesting and important.

However, the authors should test whether this particular truncation is specifically affecting IP3R1-dependent functions or also other iRhom2-mediated functions, such as ADAM17 maturation, Sting stability, or substrate selective shedding events. Also, does this mutant still interact with BCL-2? Similarly, the mutants in figure panel 5F should be tested for their capacity to interact with IP3R1 as well as with other known iRhom2 partners. Some of these interactions have already been described previously and these findings could simply be discussed in the manuscript in more detail.

The mammalian studies are currently limited to in vitro experiments with fibroblasts: the question arises as to whether these findings translate to similar functions in other cell types and if the iRhoms are important for ER stress-induced cell death in vivo or not. Since iRhom1/2-deficient mice die very early, these studies might be challenging to conduct and should not delay the publication of this significant work.

However, this novel and important mechanism should be explored in other cell types as well (e.g., iRhom2-deficient immune cells that are naturally iRhom1-deficient). In addition, this limitation and the potential physiological in vivo relevance should be discussed. Some details on how to overcome these limitations in the future should be provided.

Minor comments:

Line 145: The authors indicate that the observed phenotype was “substantially” rescued in flies with a null iRhom mutation. It would be helpful to the reader if this “substantial” rescue could be quantified based on the collected data.

Line 339: The authors should consider to provide some additional information on how the targeted BCL-2 constructs are functioning in their system. Some readers might not be familiar on how these constructs specifically target the ER or the mitochondria.

Specific bands for iRhom proteins can be detected at approximately 90-120 kDa. The authors should provide some explanation/information to the readers and discuss or speculate about the nature of the wide range of signals in some of their western blot analyses.

Reviewer #2 (Remarks to the Author):

Dulloo et al. add another pseudo-enzyme to programmed cell death by showing that iRhoms interact with IP3Rs and BCL2 to regulate calcium release from the ER during ER stress to promote cell death. These are challenging, generally well-executed experiments and the authors provide cell-based results and detailed molecular interaction studies. There are several points to resolve or clarify.

1. Given that caspase-3 activity is no longer assurance that cells are dying by apoptosis vs. other end stage mechanisms, it would be useful to know a more details about the dying process in Fig 1 where caspase activity or the consequences of its activity (e.g. Annexin V, PARP) are the primary readouts typically at 18 h, which may be relatively early in the dying process based on the extent of cell death. The authors could show a more complete time course and whether all the cells in fact die from this treatment (before media exhaustion) at least comment on the apoptotic or non-apoptotic morphology of dying cells. Whether apoptotic or not, is inconsequential as BCL2 can inhibit non-apoptotic death, only to be more rigorous.

2. While it is interesting to see the iRhom null fly as a physiological readout of ER stress, the images in Fig 1d are not high quality and could be misleading to readers (controls look less similar than ninaE-G69D with/without iRhom to untrained eyes). [It would be more interesting if Debcl, Buffy or fly IP3Rs were also involved.] Size of scale bar should be added to Fig 1d legend.

3. In Fig 1e, basal GRP94/78 levels are elevated in the iRhom DKO cells – is there underlying ER stress occurring in the absence of iRhoms as a confounder? Furthermore, statements about iRhom acting downstream of CHOP are not supported or not explained, as data appear to argue that iRhoms could work either downstream or independently of ER death pathways.

4. Add arrows and explain the multiple bands of iRhom1/2 so others can better appreciate what otherwise appears to be a messy blot (Fig 2C). More troubling is their dramatic decrease with tunicamycin treatment, making it hard to interpret their specific effects.

5. Depolarization of mitochondria can be a contributor or a consequence of programmed cell death. The authors seem to imply a contributing role, but the data do not distinguish these possibilities (Fig 2d). If the goal is to demonstrate apoptosis, then measuring apoptotic outer membrane permeabilization (distinct from PTP opening) by cytochrome c release from mitochondria would be more supportive of the claim that caspase-9 rather than caspase-8 is involved to explain lack of role for TNFR/death receptors (Fig 2e), although causal roles for caspases aren't examined. Also, the Methods were not clear: MitoTracker would give similar results as using potentiometric TMRE to assess mitochondrial inner membrane potential collapse, or does the graph represent the relative decrease in TMRE per cell? The data presented in Fig 2d also does not support the statement in the related text on p6 that iRhoms "...do not regulate the core mitochondrial machinery that controls membrane potential," or cite appropriate literature.

6. As presented there appears to be a problem with the interpretation of Fig 2e in ruling out ADAM17 in this death pathway. Given the critical role of iRhoms for the function of ADAM17 (also authors' review, Ref 7), the application of ADAM17 inhibitors to iRhom1/2 double KO cells, where ADAM17 function may already be impaired, needs further explanation and positive controls for ADAM17 function and drug inhibition. Also, the conclusion that there is enhanced casp-3 cleavage in iRhom DKO cells treated with tunicamycin plus ADAM17 inhibitor (but not with GI that inhibits both ADAM 10 and 17) is difficult to appreciate from the blots and logically, and is not quantified (Fig 2e, right lanes). The conclusion with TNF+CHX is potentially interesting (Fig 2f); if iRhom DKO cells do in fact die equally rapidly when treated with CHX+TNF (time course of cell viability not presented), perhaps consider testing CHX+TNF+zVAD to address whether iRhoms instead promote necroptosis, another death pathway downstream of TNF.

7. Model in Fig 6e, upper left, position of BCL2 is unsupported by the evidence, especially without testing the interaction between IP3R, iRhom and BCL2 in the context of ER stress in co-IP experiments.

Minor

8. Clarify the PARP western in Fig 1, which detects the N-terminal 25kD ("c-PARP") while other figures detect the standard 85 kD C-terminal fragment. However, the only anti-PARP antibody mentioned in the Methods p23 Cell Signaling #9542 does not detect the 25 kD fragment. Looking up Ref 27, which is cited for detecting "several fragments", instead reported detection of the 25 and 85 kD PARP fragments using two different antibodies.

9. Does calbryte 520 entered and become esterified equally in wt and DKO cells (Fig 3e)?

10. Fig 4a: iR2 WBs using HA or an iRhom specific antibody? Do good antibodies for iRhom exist? Everything here is with expressed, tagged iRhoms.

11. Please provide higher mag insert to visualize red puncta in Fig 4c.

12. Quantified data for Fig 4f or other evidence for reproducibility would be useful. Same applies to other blots.
13. Explain for Fig 5a: “2 independent pairs of MEFS” – independently derived MEFs, independently cultured? How many independently derived MEFs were used in this study ?
14. Fig 5b: clarify if using GFP antibody to detect IP3R1/3 or IP3R specific antibodies.
15. Fig 5c: statistics to confirm no difference in BCL2 transcript level needed; also quantify BCL2 blots. [Note SEM rather than SD plotted (Methods).]
16. S4a: why use caspase 3 as a readout for the XL knockdown and caspase 9 for the BCL2 knockdown?
17. Fig 5e: a blot for BCLxL as a negative control would be useful.
18. Fig 6a: death assay would verify cell death would be useful.
19. Is there a decrease or redistribution of IP3Rs on ER in iRhom DKO?
20. Preferable if BIRD2 in co-IP experiments to confirm that BIRD2 actually disrupts the interaction between IP3R and BCL2.
21. Adjusting the language could help with many of these criticisms.

Reviewer #3 (Remarks to the Author):

This is a well written and interesting manuscript reporting a novel role for iRhom pseudoproteases in regulating endoplasmic reticulum (ER) stress-induced cell death. The major findings are as follows. The authors show that mouse embryonic fibroblasts (MEFs) genetically deficient in iRhom1/2 (DKO) are strongly resistant to apoptosis in response to pharmacologic inducers of ER stress, including tunicamycin and brefeldin A. Similarly, Drosophila mutants in the single iRhom gene are partly protected against photoreceptor cell loss due to a mutation in rhodopsin 1 that cripples its folding. The authors provide evidence that these ER stress induces activate the proximal unfolded protein response (UPR) components (IRE1, PERK, ATF6) normally in the DKO cells, but fail to induce mitochondrial depolarization and downstream Caspase activation. Given that ER Ca²⁺ release followed by mitochondrial Ca²⁺ uptake is an important contributor to ER stress-induced apoptosis, they examined the role of iRhom1/2 in this process. They show that iRhom 1/2 interact with IP3 receptors at the ER membrane as well as the anti-apoptotic BCL2 protein, which is known to bind IP3 receptors and inhibit Ca²⁺ release. They provide evidence that iRhom 1/2 attenuate the inhibitory

interaction between BCL-2 and IP3 receptors, and that this is the mechanism by which they regulate ER stress-induced apoptosis.

Overall, the data are well controlled, comprehensive and compelling. Their findings represent an important discovery that should be of broad interest to the cell death and ER stress fields.

However, there are several issues that should be addressed to clarify their data and improve the manuscript:

1) To better understand whether the DKO are only resistant to ER stress, they should challenge these cells with a range of non-ER stress stimuli (such as staurosporine, etoposide, etc) that trigger mitochondrial-dependent apoptosis.

2) When examining the activation of the UPR in the wt vs DKO cells (Fig 1E-F), they should do a dose response and time course of tunicamycin to see if there are any subtle defects in its activation.

3) Considering that thapsigargin causes ER Ca²⁺ release by blocking the SERCA pump, it is odd that they never use this drug in the study. They should test it as another ER stress agent (along with tunicamycin) and see if the DKO are as resistant to it as tunicamycin and BFA. Likewise, they should also use thapsigargin in Figure 3 as a way to release total ER Ca²⁺ stores and measure cytosolic Ca²⁺ uptake (to see if there is any difference in stores between the wt and DKO cells).

4) They should examine BID (cleavage) and BIM (levels and phosphorylation) in the DKO cells upon ER stress to rule out a defect at the level of BH3-only proteins.

5) They should test if the ER-localized version of BCL-2 is as effective at rescuing the ER Ca²⁺ defects and ER stress-induced apoptosis in the DKO cells as wild-type BCL-2.

Reviewer #4 (Remarks to the Author):

The study of Dulloo et al highlights a novel function of iRhom pseudoproteases in ER stress. The authors demonstrate that knocking out the iRhoms protects against ER stress and propose a

mechanism by which these pseudoproteases interact with IP3 receptors and Bcl-2 to regulate ER/mitochondrial Ca²⁺ signaling and cell death. The paper is clearly written and the findings are both important and novel. However, the present study lacks direct evidence that iRhoms have functional effects on IP3Rs and insights into the mechanism by which ER stress regulates the IP3R/Bcl2/iRhoms complex. In addition, there is no discussion of possible caveats in the use of severely mutated constructs and there are specific problems in interpreting some of the blotting data. These concerns are detailed below.

DETAILED COMMENTS

1) Figure 1D shows experiments on *Drosophila* mutants in which the iRHOM gene knockout is indicated as “substantially rescuing” the phenotype of a *ninaE* mutation that causes ER stress in the eye. However, the figure panels on the left and the quantitation on the right indicates that the iRhom^{-/-} has effects on its own and has only a limited effect in rescuing the *ninaE* phenotype since most of the photoreceptors are still abnormal in the double mutant.

2) The section of the text referring to Figure 1 panels e & f are confusing. The data in the panels show that none of the initiating steps of the UPR are affected in the iRhom Kos. This includes RT-PCR measurements of a CHOP target gene *Ero1L*. Yet, the final sentence (L162) concludes that the effects of iRhom deletion must be downstream of CHOP.

3) What is the distinction between the PARP cleavage being examined in Figure 2b where the fragment shown is at ~80kDa and the PARP cleavage in Figure 1c where the fragment shown is at ~25kDa. In Fig2c the authors carry out rescue experiments in which the reconstitution of iRhom1 & 2 is proposed to restore caspase-3 and PARP cleavage “almost to WT levels”. In the case of PARP it appears to be more than WT. However, the blots do not show the precursor bands for PARP although the bands were shown in the blot of panel 2b. Proper interpretation of these experiments requires the precursor bands to be shown.

4) Why does inhibition of ADAM17 increase caspase-3 cleavage? (Figure 2e)

5) The sentence on p7 reads: “iRhoms are substantially localized in the ER, so we investigated how they could affect mitochondrial membrane potential under ER stress”. The sentence is a bit confusing because the membrane potential was already measured in the previous figure and this section of the MS deals with a Figure measuring Ca²⁺.

6) The data in Figure 3 is central to the contention that iRhoms are regulators of IP3R-mediated Ca²⁺ release. Yet, all the data are measurements of cytosolic Ca²⁺ changes induced by agonists in which the effects of iRhoms could be on many steps which precede IP3 formation. Measurements using permeabilized cells or caged IP3 should be used to confirm that iRhoms have direct effects on IP3R function.

7) 2-APB is not a specific inhibitor of IP3Rs and has major effects on Ca²⁺ entry channels. I think the authors should mention this as a caveat in the description of Figure 3f.

8) Why is there so much cleavage of the mutant construct used in Fig 4e? The decreased binding to IP3R1 may be due to inefficient targeting of the mutant to the ER. The same problem may apply to the interpretation of the data with N-terminal mutants in Fig 5f.

9) One feature of this study is that different end-points are used for ER stress even in the same series of experiments. For example caspase-3 effects are shown in Fig5b and caspase-9 effects are shown in Figure 5d. Is there a reason for this?

10) In Figure 5d the authors conclude that knockdown of Bcl2 abolished the suppression of tunicamycin ER stress in iRhom2 KO cells based on the intensity of the caspase-9 cleaved band. However, the intensity of this band is dependent on the abundance of its precursor which appears higher in the Bcl-2 KO cells. In any case, some kind of quantitation which takes this into account is necessary to back up this conclusion.

11) A key part of the model shown in Fig 6e is that ER stress dissociates the iRhom/Bcl2 complex from IP3Rs and permits mitochondrial Ca²⁺ overload. However, the paper has no experiments (or speculations) on why this dissociation occurs.

Dulloo et al. NCOMMS-20-47103A

Response to reviewers' comments

We are grateful to the four reviewers for their detailed and constructive comments. We are of course very pleased that all were positive, commenting specifically on the novelty and quality of our approaches. They have also challenged us with a number of suggestions and specific criticisms that we respond to in detail below.

Reviewer #1 (Remarks to the Author):

iRhoms have emerged as key regulators of ADAM17-dependent TNF activation and their role in controlling ADAM17-mediated EGFR-signaling is well-established. However, although iRhoms have recently been associated with other biological functions, their exact roles in this regard have only recently begun to be elucidated in a limited number of studies. In the current study, Dulloo et al. present compelling biochemical and cell biological evidence for the involvement of two catalytically inactive members of rhomboid proteases in ER stress-induced cell death through IP3 receptors and BCL-2 signaling pathways. The authors show that both iRhoms are necessary to regulate ER stress-induced calcium influx into mitochondria and that loss of iRhoms causes cells to be resistant to ER stress-induced apoptosis. They convincingly demonstrate that iRhoms are needed for the activation of apoptotic caspases, specifically via the intrinsic mitochondrial cell death pathway. The authors deploy a wide range of elegant techniques to assess the role of these two inactive rhomboids in controlling apoptosis due to persistent ER stress.

The manuscript is well written and the observation that iRhoms control ER stress-induced cell death is novel. I only have a few comments that the authors should address in their revised manuscript.

We are very pleased that this Reviewer is so broadly supportive of our work.

Introduction:

Line 63: Previous studies have provided evidence for the location of iRhoms in the endoplasmic reticulum. The authors should specify in their introduction whether these observations were based on overexpression experiments or detection of endogenous protein in mammalian cells and/or drosophila.

Text added.

Results:

While the quality of the majority of western blots is excellent, especially when considering the challenges that are associated with Co-IPs, the data presentation in figure panel 2c is less convincing. Similarly, to figure panel 2B, the "full-length" PARP should be shown as well. Also, the differences in cleaved caspase 3 levels are difficult to estimate in this particular blot and could potentially be replaced with another representative image. Alternatively, quantifications for figure panels with whole cell lysates could be provided.

We have replaced Fig. 2C with another blot to show full-length and cleaved forms of both PARP and caspase-3.

Several structure/function studies have explored the role of individual segments of these multi-membrane spanning proteins. The authors' observations that a truncated mutant containing only the cytoplasmic N-terminal domain, the first transmembrane domain as well as the luminal domain showed significantly reduced binding with IP3R1 is interesting and important. However, the authors should test whether this particular truncation is specifically

affecting IP3R1-dependent functions or also other iRhom2-mediated functions, such as ADAM17 maturation, Sting stability, or substrate selective shedding events.

The reviewer is right to highlight the modular function of the different segments of iRhom proteins. We have now investigated the effect of this mutant (iR2_TMD1_IRHD) on two previously studied functions of iRhom2, namely ADAM17 maturation and EGF ligand stability. Both wild type iR2 and mutant iR2_TMD1_IRHD bind and degrade EGF protein equally (Suppl. Fig.4d). Conversely, mutant iR2_TMD1_IRHD was unable to bind and induce ADAM17 maturation (Suppl. Fig.4e), indicating that the TMD2->TMD7 of iRhom2 is essential for this function. Although we include this data in supplementary figures for completeness, its interpretation is difficult because ADAM17 is a single-pass transmembrane protein and IP₃R1 has multiple TMDs. Ultimately, we may need structural information to understand fully whether the interaction of these two clients with iRhom2 uses the same interface.

Also, does this mutant still interact with BCL-2?

We show in Fig. 5f, that the interaction of cytoplasmic BCL-2 protein primarily occurs via the N-terminal cytoplasmic domain of iRhom2, which is still present in the mutant iR2_TMD1_IRHD.

Similarly, the mutants in figure panel 5F should be tested for their capacity to interact with IP3R1 as well as with other known iRhom2 partners.

The iR2 mutants used in Fig. 5f are serially truncated within the N-terminal cytoplasmic domain, with the remaining parts of the protein intact. We previously show in Fig. 4f that the N-terminal cytoplasmic domain of iRhom2 does not contribute to its binding to IP₃R1.

Some of these interactions have already been described previously and these findings could simply be discussed in the manuscript in more detail.

Agreed. As suggested, we have tried to make this clearer and more streamlined in text. We believe this addresses both the two preceding points.

The mammalian studies are currently limited to in vitro experiments with fibroblasts: the question arises as to whether these findings translate to similar functions in other cell types and if the iRhoms are important for ER stress-induced cell death in vivo or not. Since iRhom1/2-deficient mice die very early, these studies might be challenging to conduct and should not delay the publication of this significant work.

We appreciate the reviewer's understanding of the challenges presented by embryonic lethal iRhom1/2-deficient mice. It is for that reason that we have used the *Drosophila* iRhom knockout flies, which are viable, and evidenced the biological relevance in *Drosophila* eye photoreceptors of this new function of iRhoms in ER stress-induced cell death (Fig. 1d).

However, this novel and important mechanism should be explored in other cell types as well (e.g., iRhom2-deficient immune cells that are naturally iRhom1-deficient).

We have used several cell types in this study to show the role of iRhoms in regulating ER stress-induced cell death. These include mouse embryonic fibroblasts (MEFs), A549 human adenocarcinoma alveolar basal epithelial cells, and *Drosophila*. As suggested by the reviewer, we have also now analysed the effect of ER stress using tunicamycin on bone marrow derived macrophages (BMDMs) cells from iRhom2-knockout mice, which are naturally iRhom1-deficient. However, these primary cells, which cannot be passaged or kept in culture for long, are intrinsically extremely sensitive to even low doses tunicamycin, and

we were unable to generate conclusive data. As an alternative, we looked at the role of iRhoms in ER stress-induced cell death in BV2a mouse microglia cells, which have low expression of iRhom1. Knockout of iRhom2 in BV2a cells, significantly inhibited PARP cleavage upon tunicamycin treatment (Suppl. Fig 1c), consistent with what we observe in other cell types tested. Overall, we feel that this variety of cell types and approaches provides good evidence that the role of iRhoms in regulating ER stress-induced cell death is widespread.

In addition, this limitation and the potential physiological in vivo relevance should be discussed. Some details on how to overcome these limitations in the future should be provided.

We have amended the Discussion as suggested to highlight the limitation and potential of this newly discovered function of iRhoms.

Minor comments:

Line 145: The authors indicate that the observed phenotype was “substantially” rescued in flies with a null iRhom mutation. It would be helpful to the reader if this “substantial” rescue could be quantified based on the collected data.

This point is also highlighted by Reviewer 4. We have replaced the panels with better EM images that make the rescue easier to see. The quantification shown in the bar chart of Fig. 1d illustrates a significant increase in the number of normal photoreceptors in *iRhom*^{-/-}; *ninaE*^{G69D} flies compared to *ninaE*^{G69D}. Note also that *ninaE*^{G69D} flies have lost about 90% of all photoreceptors, whereas *iRhom*^{-/-}; *ninaE*^{G69D} flies have about 75% of wild-type numbers of photoreceptors, albeit many of them with abnormal morphology. Nevertheless, we have amended the text to remove the rather vague use of 'substantially'.

Line 339: The authors should consider to provide some additional information on how the targeted BCL-2 constructs are functioning in their system. Some readers might not be familiar on how these constructs specifically target the ER or the mitochondria.

Agreed. We have now included more details about the targeted BCL-2 constructs. Specific bands for iRhom proteins can be detected at approximately 90-120 kDa. The authors should provide some explanation/information to the readers and discuss or speculate about the nature of the wide range of signals in some of their western blot analyses.

We thank the referee for highlighting these discrepancies in the sizing of iR2 protein in some blots. Full length iR2 protein runs at around 95kDa. As indicated in the Methods and manufacturer's protocol, the same protein marker used throughout the paper runs at slightly different molecular weights, depending on the type of SDS-PAGE used (Tris-Glycine v/s Bis-Tris gels), as per manufacturer's description. This led to the inadvertent misalignment and mislabelling of some protein size markers. We have made the necessary corrections.

Reviewer #2 (Remarks to the Author):

Dulloo et al. add another pseudo-enzyme to programmed cell death by showing that iRhoms interact with IP3Rs and BCL-2 to regulate calcium release from the ER during ER stress to promote cell death. These are challenging, generally well-executed experiments and the authors provide cell-based results and detailed molecular interaction studies.

Thank you for these positive comments.

There are several points to resolve or clarify.

1. Given that caspase-3 activity is no longer assurance that cells are dying by apoptosis vs. other end stage mechanisms, it would be useful to know a more details about the dying process in Fig 1 where caspase activity or the consequences of its activity (e.g. Annexin V, PARP) are the primary readouts typically at 18 h, which may be relatively early in the dying process based on the extent of cell death.

The authors could show a more complete time course and whether all the cells in fact die from this treatment (before media exhaustion) at least comment on the apoptotic or non-apoptotic morphology of dying cells. Whether apoptotic or not, is inconsequential as BCL-2 can inhibit non-apoptotic death, only to be more rigorous.

Our data show cleaved PARP at 18hrs, a well-defined marker of end-stage apoptosis, supporting the proposal that the cell death indicated by the PI/Annexin V assay is caused at least primarily by apoptosis. We do, however, agree with the reviewer that a time course of cell death after ER stress induction is a useful addition to the data and we have now included this in supplementary Fig. 1a. Tunicamycin treatment of up to 36hrs led to [>90%] death of wild-type cells; iRhom1/2 DKO cells are still significantly resistant [\pm 60% death], but do eventually die after continued exposure to tunicamycin, presumably due to eventual non-specific activation of apoptotic pathways and/or by non-apoptotic pathways. We have also now added morphological description of the dying cells upon tunicamycin treatment at 18hrs in the results section, pointing out that the process of death resembles apoptosis (for example reduced cell volume and fragmented nuclei).

2. While it is interesting to see the iRhom null fly as a physiological readout of ER stress, the images in Fig 1d are not high quality and could be misleading to readers (controls look less similar than *ninaE-G69D* with/without iRhom to untrained eyes). [It would be more interesting if *Debcl*, *Buffy* or fly IP₃Rs were also involved.] Size of scale bar should be added to Fig 1d legend.

Agreed. We have now provided a better quality picture of the WT control, and have also added a diagram of an ommatidium to clarify and to facilitate the data interpretation. The size of scale bar has been added in the legend.

While we agree that looking in more detail at the genetics of cell death in this fly model, for example with *Debcl*, a pro-apoptotic member of the Bcl-2 family, *Buffy* which encodes the other Bcl-2 family member, or IP₃Rs, we see that as a new, *Drosophila*-based study and therefore outside the current scope.

3. In Fig 1e, basal GRP94/78 levels are elevated in the iRhom DKO cells – is there underlying ER stress occurring in the absence of iRhoms as a confounder?

There are no significant differences in basal ER stress markers GRP94/GRP78: the apparently slightly elevated level in Fig. 1e is not representative. These proteins have similar levels at both steady- and induced-state, in wild-type and iRhom1/2 double knockout cells, as shown in Fig. 1f, Fig. 2b, Fig. 5c, Fig. 6a, Suppl. Fig. 1d, and Suppl. Fig. 2a. We have now replaced the GRP94/GRP78 in Fig. 1e with a more representative blot.

Furthermore, statements about iRhom acting downstream of CHOP are not supported or not explained, as data appear to argue that iRhoms could work either downstream or independently of ER death pathways.

Thank you for making this point, which was also highlighted by Reviewer 4. We agree and have rewritten the relevant text, acknowledging that the alternative conclusion that iRhoms could be acting independently of CHOP.

4. Add arrows and explain the multiple bands of iRhom1/2 so others can better appreciate what otherwise appears to be a messy blot (Fig 2C). More troubling is their dramatic decrease with tunicamycin treatment, making it hard to interpret their specific effects.

In response to the comments of this reviewer and reviewer 4, we have replaced Fig. 2c with a better quality blot. It is well established by many groups that iRhoms express as a full-length protein of about 100kDa and a major cleaved form at around 50kDa, when tagged at the C-terminal. Depending on the type of expression (endogenous, stable, transient), extra bands are also often observed, likely due to further proteolytic cleavage of the full-length protein. We have added arrows to mark those major bands and other cleaved forms.

iRhoms are glycosylated ER membrane proteins, making their folding and stability sensitive to tunicamycin, which inhibits protein glycosylation. In these cells, the level of iRhoms is sufficient to rescue the apoptotic defect of iRhom1/2 DKO cells, despite being low and reduced by tunicamycin. The reviewer is correct that these effects must be taken into account, but we believe the interpretations are rigorous.

5. Depolarization of mitochondria can be a contributor or a consequence of programmed cell death. The authors seem to imply a contributing role, but the data do not distinguish these possibilities (Fig 2d). If the goal is to demonstrate apoptosis, then measuring apoptotic outer membrane permeabilization (distinct from PTP opening) by cytochrome c release from mitochondria would be more supportive of the claim that caspase-9 rather than caspase-8 is involved to explain lack of role for TNFR/death receptors (Fig 2e), although causal roles for caspases aren't examined.

In response to this suggestion, we have now looked at the release of cytochrome c from the mitochondria into the cytoplasm, using cell fractionation and western blot. iRhom1/2 DKO cells show a lower level of cytochrome c release in the cytoplasm compared to wild type cells after tunicamycin treatment (Suppl. Fig 2c), further supporting the role iRhoms in mitochondrial mediated apoptosis.

Also, the Methods were not clear: MitoTracker would give similar results as using potentiometric TMRE to assess mitochondrial inner membrane potential collapse, or does the graph represent the relative decrease in TMRE per cell? The data presented in Fig 2d also does not support the statement in the related text on p6 that iRhoms "...do not regulate the core mitochondrial machinery that controls membrane potential," or cite appropriate literature.

The graph represents the decrease in TMRE fluorescence after each treatment, calculated relative to untreated cell populations. We have added more details in the Methods section to clarify this.

We agree with the reviewer that the point about iRhoms not regulating the core mitochondrial machinery controlling membrane potential is not justified and we have removed it.

6. As presented there appears to be a problem with the interpretation of Fig 2e in ruling out ADAM17 in this death pathway. Given the critical role of iRhoms for the function of ADAM17 (also authors' review, Ref 7), the application of ADAM17 inhibitors to iRhom1/2 double KO cells, where ADAM17 function may already be impaired, needs further explanation and positive controls for ADAM17 function and drug inhibition.

The reviewer is correct that iRhoms are critical regulators of ADAM17 and therefore in the iRhom1/2 DKO cells, ADAM17 processing and activity are inhibited. We included this fairly predictable and minor result for the sake of completeness, with the aim of emphasising that the only well characterised function of iRhoms – regulating ADAM17 – is not relevant here. We are happy to take editorial advice about whether this should be removed. If retained, we now also include positive controls for the ADAM17 inhibitors used, showing that ADAM17-dependent release of AREG is blocked by the inhibitors (Suppl. Fig 2c).

Also, the conclusion that there is enhanced casp-3 cleavage in iRhom DKO cells treated with tunicamycin plus ADAM17 inhibitor (but not with GI that inhibits both ADAM 10 and 17) is difficult to appreciate from the blots and logically, and is not quantified (Fig 2e, right lanes).

In response to this and a similar comment from Reviewer 4 comments, we have now quantified the cleaved caspase-3 bands and concluded there is no significant difference when treated with tunicamycin+ADAM17 inhibitors. We have also replaced Fig. 2e with a more representative blot.

The conclusion with TNF+CHX is potentially interesting (Fig 2f); if iRhom DKO cells do in fact die equally rapidly when treated with CHX+TNF (time course of cell viability not presented), perhaps consider testing CHX+TNF+zVAD to address whether iRhoms instead promote necroptosis, another death pathway downstream of TNF.

We agree that investigating the possibility that iRhoms promote necroptosis downstream of TNF signalling would be interesting to pursue but believe that this would be the basis for a future piece of work and is beyond the scope of this paper, which aims to set the scene for the role of iRhoms in specifically apoptotic cell death.

7. Model in Fig 6e, upper left, position of BCL-2 is unsupported by the evidence, especially without testing the interaction between IP₃R, iRhom and BCL-2 in the context of ER stress in co-IP experiments.

We have struggled with how best to represent our model in this figure and have now amended the original model in Fig. 6e. The model is based on the following characteristics. (1) Our data shows the existence of a regulatory complex consisting of iRhom2, IP₃Rs and BCL-2 (Fig. 5e); each protein interacts with the other, but we have not rigorously proved that this there is a tripartite complex where they all interact together at the same time. (2) The interaction of BCL-2 at the coupling domain in the cytoplasmic part of IP₃Rs is well documented (Ref. 49,52,54). (3) Our data show iRhom2 interacts with BCL-2 via the cytoplasmic domain of iRhom2 (Fig. 5f), and with IP₃Rs via the TMDs (Fig. 4d-e). (4) The latter interaction is further enhanced under ER stress conditions (Fig. 4f). (5) BCL-2 dissociates from IP₃R1 in the presence of iRhom2 (Fig. 6c-d). Finally, (6), we now include data showing no effect of ER stress on BCL-2 binding to iRhom2 (Suppl. Fig 6c).

Minor

8. Clarify the PARP western in Fig 1, which detects the N-terminal 25kD (“c-PARP”) while other figures detect the standard 85 kD C-terminal fragment. However, the only anti-PARP antibody mentioned in the Methods p23 Cell Signaling #9542 does not detect the 25 kD fragment. Looking up Ref 27, which is cited for detecting “several fragments”, instead reported detection of the 25 and 85 kD PARP fragments using two different antibodies.

Also raised by Reviewer 4, we have now made the PARP blots consistent throughout the paper. We have replaced Fig. 1c with a blot showing uncleaved PARP (116 KDa), and cleaved PARP (86 KDa and 24kDa). Note that the PARP antibody (Cell Signaling, #9542)

used throughout the study, does in fact detect the smaller cleaved product of 24 kDa, as shown in Fig. 1c. The company's data sheet for this antibody only shows bands up to 57kDa and therefore misses the smaller band.

9. Does calbryte 520 enter and become esterified equally in wt and DKO cells (Fig 3e)?

We have now investigated the amount of de-esterified Calbryte-520 in both cell lines and it is indistinguishable in WT and iRhom1/2 DKO MEFs as shown in Suppl. Fig 3c.

10. Fig 4a: iR2 WBs using HA or an iRhom specific antibody? Do good antibodies for iRhom exist? Everything here is with expressed, tagged iRhoms.

There are no reliable antibodies for the detection of endogenous iRhoms for western blots. We and other groups studying iRhoms primarily express tagged iRhoms. Wherever possible, we use stable cell lines of iRhoms, with relatively low expression. All interactions are confirmed with both exogenously and endogenously expressed binding proteins.

11. Please provide higher mag insert to visualize red puncta in Fig 4c.

Done.

12. Quantified data for Fig 4f or other evidence for reproducibility would be useful. Same applies to other blots.

We have now replaced this figure with a better quality blot and have also included the quantification of Fig. 4f and also of other selected blots throughout the paper.

13. Explain for Fig 5a: "2 independent pairs of MEFs" – independently derived MEFs, independently cultured? How many independently derived MEFs were used in this study?

The two independent pairs of MEFs used in Fig. 5a were derived from different mouse embryos and we have now included this in the text.

14. Fig 5b: clarify if using GFP antibody to detect IP3R1/3 or IP3R specific antibodies.

We have now clarified the use of GFP antibody to detect IP₃R1 and IP₃R3 by including GFP in the labelling of the blot.

15. Fig 5c: statistics to confirm no difference in BCL-2 transcript level needed; also quantify BCL-2 blots. [Note SEM rather than SD plotted (Methods).]

We have added the statistics for the qPCR data and included a quantification of the western blots, with relevant description inserted in the text

16. S4a: why use caspase 3 as a readout for the XL knockdown and caspase 9 for the BCL-2 knockdown?

To be more consistent throughout Fig. 5, we have replaced the blot in Fig. 5d (siBCL2) with one showing caspase-3, similar to the readout shown in new Suppl. Fig. 5a (siBCL-XL) and Fig. 5b.

17. Fig 5e: a blot for BCLxL as a negative control would be useful.

We have now included the blot for BCL-XL in the co-IP experiment.

18. Fig 6a: death assay would verify cell death would be useful.

We take the point but at 18 hrs of treatment, which is what we used for this blot, although there is a clear rescue of caspase-3 cleavage by BIRD-2, the late markers of cell death (e.g. binding to phosphatidylserine on the surface of apoptotic cells) required for detection by PI/Annexin V cell death assay, are not yet distinguishable between treated and untreated cells. At later times, cell death quantification becomes difficult because all tunicamycin-treated wild type cells, which act as the positive controls for the rescue of iRhom1/2 DKO cells with BIRD-2+tunicamycin, have died by then. These complications make us prefer to use the rescue of cleavage of apoptotic caspase-3 as the indicator of rescue by BIRD-2.

19. Is there a decrease or redistribution of IP₃Rs on ER in iRhom DKO cells?

As shown in Fig. 5a, there is an increase in IP₃R1 and a decrease in IP₃R3 protein levels in iRhom1/2 DKO cells. To explore whether these changes might account for the altered resistance to cell death, we previously overexpressed IP₃R1 and IP₃R3 in these cells; there was no change their resistance to ER stress-induced cell death (Fig. 5b), implying that expression levels of IP₃Rs are not the determining factors. We have rewritten the relevant text and hope it is clear. We have tried hard to examine the distribution of IP₃Rs in iRhom1/2 DKO cells but immunostaining of IP₃R1 and IP₃R3 did not work well in the MEFs and we cannot draw a firm conclusion.

20. Preferable if BIRD2 in co-IP experiments to confirm that BIRD2 actually disrupts the interaction between IP₃R and BCL-2.

The effect of BIRD-2 effect in disrupting the interaction between IP₃Rs and BCL-2 has already been well characterised (Ref. 57-62). We have now also validated it ourselves and show its inhibitory effect on the binding BCL-2 and IP₃R1 (Suppl. Fig. 6a).

21. Adjusting the language could help with many of these criticisms.

We much appreciate the thoughtful and comprehensive advice from Reviewer 2 and have made many textual adjustments to try and capture these points and make the arguments easier to follow.

Reviewer #3 (Remarks to the Author):

This is a well written and interesting manuscript reporting a novel role for iRhom pseudoproteases in regulating endoplasmic reticulum (ER) stress-induced cell death. The major findings are as follows. The authors show that mouse embryonic fibroblasts (MEFs) genetically deficient in iRhom1/2 (DKO) are strongly resistant to apoptosis in response to pharmacologic inducers of ER stress, including tunicamycin and brefeldin A. Similarly, Drosophila mutants in the single iRhom gene are partly protected against photoreceptor cell loss due to a mutation in rhodopsin 1 that cripples its folding. The authors provide evidence that these ER stress induces activate the proximal unfolded protein response (UPR) components (IRE1, PERK, ATF6) normally in the DKO cells, but fail to induce mitochondrial depolarization and downstream Caspase activation. Given that ER Ca²⁺ release followed by mitochondrial Ca²⁺ uptake is an important contributor to ER stress-induced apoptosis, they examined the role of iRhom1/2 in this process. They show that iRhom 1/2 interact with IP₃ receptors at the ER membrane as well as the anti-apoptotic BCL-2 protein, which is known to bind IP₃ receptors and inhibit Ca²⁺ release. They provide evidence that iRhom 1/2 attenuate the inhibitory interaction between BCL-2 and IP₃ receptors, and that this is the mechanism by which they regulate ER stress-induced apoptosis.

Overall, the data are well controlled, comprehensive and compelling. Their findings represent an important discovery that should be of broad interest to the cell death and ER stress fields.

We are of course very pleased with this overall assessment of our manuscript.

However, there are several issues that should be addressed to clarify their data and improve the manuscript:

1) To better understand whether the DKO are only resistant to ER stress, they should challenge these cells with a range of non-ER stress stimuli (such as staurosporine, etoposide, etc) that trigger mitochondrial-dependent apoptosis.

As requested, we have now tested the effect of etoposide, which triggers mitochondrial dependent apoptosis by genotoxic stress. iRhom1/2 DKO cells are also resistant to this as shown by PARP, caspase-3 and caspase-9 cleavage (Suppl. Fig. 2a). Although we have not followed this up, it is noteworthy that other stresses can lead to Ca^{2+} -mediated apoptosis (Ref. 33-35), so this result is consistent with our conclusion that iRhoms are integral to stress-induced apoptosis via the mitochondrial cell death pathway.

2) When examining the activation of the UPR in the wt vs DKO cells (Fig 1E-F), they should do a dose response and time course of tunicamycin to see if there are any subtle defects in its activation.

We agree with the reviewer about the value of a dosage and time course of the effect of tunicamycin on wild type and iRhom1/2 DKO cells. In Fig. 1f, we previously presented a time course of tunicamycin (0h, 3h, 6h, 9h, 16h), showing no significant difference for *Xbp1* splicing, ATF6 cleavage, or GRP94/GRP78 induction. We have now also added a dose response of tunicamycin (DMSO, 0.25, 0.5, 1, 2 μ g/ml), which also shows no difference in UPR markers (Suppl. Fig. 1d).

3) Considering that thapsigargin causes ER Ca^{2+} release by blocking the SERCA pump, it is odd that they never use this drug in the study. They should test it as another ER stress agent (along with tunicamycin) and see if the DKO are as resistant to it as tunicamycin and BFA.

Good point. We have now added data to show that iRhom1/2 DKO cells are resistant to thapsigargin-induced cell death as indicated by reduction of PARP, caspase-3 and caspase-9 cleavage (Suppl. Fig. 2a).

Likewise, they should also use thapsigargin in Figure 3 as a way to release total ER Ca^{2+} stores and measure cytosolic Ca^{2+} uptake (to see if there is any difference in stores between the wt and DKO cells).

In response to this suggestion, we have now used thapsigargin to determine ER Ca^{2+} content. As previously observed with ionomycin (Suppl. Fig. 3b), there is no significant difference between wild type and iRhom1/2 DKO cells ER Ca^{2+} stores (new Fig. 3e).

4) They should examine BID (cleavage) and BIM (levels and phosphorylation) in the DKO cells upon ER stress to rule out a defect at the level of BH3-only proteins.

It's an interesting question and, as suggested, we examined BID and BIM proteins, both pro-apoptotic BH3-only activators, which induce pore formation and mitochondria outer membrane permeabilisation. We detected very slight cleavage of BID in wild type at 18hrs and 24hrs after tunicamycin exposure, but not in iRhom1/2 DKO cells. Conversely, total and

phosphorylated BIM levels were slightly elevated in iRhom1/2 DKO cells compared to wild-type cells. These minor effects we observe, which are likely to compensate for each other functionally (Ren, D. *et al.* 2010, *Science* 330), seem unlikely to have any major contribution to the apoptotic defect observed in iRhom1/2 DKO cells in response to tunicamycin. We therefore prefer not to include these essentially negative results, but we are happy to take editorial advice if they should be included as a supplementary figure.

5) They should test if the ER-localized version of BCL-2 is as effective at rescuing the ER Ca²⁺ defects and ER stress-induced apoptosis in the DKO cells as wild-type BCL-2.

Actually, BCL-2 is an inhibitor of IP₃R function so neither wild-type nor ER tethered forms is expected to rescue ER calcium defects or stress induced apoptosis. Apologies if our text was confusing on this point, we have tried to improve its clarity.

Reviewer #4 (Remarks to the Author):

The study of Dulloo et al highlights a novel function of iRhom pseudoproteases in ER stress. The authors demonstrate that knocking out the iRhoms protects against ER stress and propose a mechanism by which these pseudoproteases interact with IP₃ receptors and Bcl-2 to regulate ER/mitochondrial Ca²⁺ signaling and cell death. The paper is clearly written and the findings are both important and novel. However, the present study lacks direct evidence that iRhoms have functional effects on IP₃R and insights into the mechanism by which ER stress regulates the IP₃R/BCL-2/iRhoms complex. In addition, there is no discussion of possible caveats in the use of severely mutated constructs and there are specific problems in interpreting some of the blotting data. These concerns are detailed below.

We appreciate the recognition of the novelty and importance of our results and hope the points below can address this reviewer's concerns.

DETAILED COMMENTS

1) Figure 1D shows experiments on *Drosophila* mutants in which the iRHOM gene knockout is indicated as “substantially rescuing” the phenotype of a *ninaE* mutation that causes ER stress in the eye. However, the figure panels on the left and the quantitation on the right indicates that the iRhom^{-/-} has effects on its own and has only a limited effect in rescuing the *ninaE* phenotype since most of the photoreceptors are still abnormal in the double mutant.

A similar point was made by Reviewer 1 and we have tried to improve these data. The EM picture has now been replaced with a higher quality and more representative image. The quantifications in Fig. 1d make it clear that loss of iRhom alone has very little effect on photoreceptors in 4-week old flies. Loss of iRhom in the context of the *ninaE*^{G69D} mutation, however, has a powerful rescuing effect. This is indicated by comparing both the total numbers of photoreceptors (of normal or abnormal morphology) detectable in *ninaE*^{G69D} compared to *iRhom*^{-/-}; *ninaE*^{G69D} flies; also by the relative number of photoreceptors with normal morphology. We have adjusted the text to remove the rather vague description of 'substantial' rescue.

2) The section of the text referring to Figure 1 panels e & f are confusing. The data in the panels show that none of the initiating steps of the UPR are affected in the iRhom Kos. This includes RT-PCR measurements of a CHOP target gene *Ero1L*. Yet, the final sentence (L162) concludes that the effects of iRhom deletion must be downstream of CHOP.

We agree with this point, which was also made by Reviewer 2. We have rewritten this statement to acknowledge the alternative possibility that iRhoms could be acting independently of CHOP.

3) What is the distinction between the PARP cleavage being examined in Figure 2b where the fragment shown is at ~80kDa and the PARP cleavage in Figure 1c where the fragment shown is at ~25kDa. In Fig2c the authors carry out rescue experiments in which the reconstitution of iRhom1 & 2 is proposed to restore caspase-3 and PARP cleavage “almost to WT levels”. In the case of PARP it appears to be more than WT. However, the blots do not show the precursor bands for PARP although the bands were shown in the blot of panel 2b. Proper interpretation of these experiments requires the precursor bands to be shown.

Another point also raised by Reviewer 2. We have now made the PARP blots consistent throughout the paper by replacing Fig. 1c with a blot showing uncleaved PARP (116 KDa), and cleaved PARP (86 KDa and 24kDa). Note also that the PARP antibody (Cell Signaling, #9542) used throughout the study, does in fact detect the smaller cleaved product of 24 kDa, as shown in Fig. 1c. The company’s data sheet for this antibody only shows bands up to 57KDa and therefore misses the smaller band.

4) Why does inhibition of ADAM17 increase caspase-3 cleavage? (Figure 2e)

Having now quantified these data, as requested by Reviewer 2 we conclude that there is no change in caspase-3 cleavage. As well as adding the quantification, we have replaced the figure with a more representative blot.

5) The sentence on p7 reads: “iRhoms are substantially localized in the ER, so we investigated how they could affect mitochondrial membrane potential under ER stress”. The sentence is a bit confusing because the membrane potential was already measured in the previous figure and this section of the MS deals with a Figure measuring Ca²⁺.

We agree and have now reordered this section, which we hope makes our points more logically ordered and clearer.

6) The data in Figure 3 is central to the contention that iRhoms are regulators of IP₃R-mediated Ca²⁺ release. Yet, all the data are measurements of cytosolic Ca²⁺ changes induced by agonists in which the effects of iRhoms could be on many steps which precede IP₃ formation. Measurements using permeabilized cells or caged IP₃ should be used to confirm that iRhoms have direct effects on IP₃R function.

The data in Fig. 3 illustrates how iRhoms regulate ER stress-induced cell death, by affecting the release of ER Ca²⁺ through IP₃Rs. We agree with the reviewer’s comment about the interpretation of the agonist-induced Ca²⁺ release data and the desirability of using other, more direct assays of IP₃R function. As suggested, we measured IP₃R-mediated Ca²⁺ release in response to IP₃ in permeabilised cells. The result was that, consistent with the agonist-induced assays, iRhom1/2 DKO cells were less responsive than wild-type cells to IP₃ (new Fig. 3d). We note, however, that the loss of responsiveness is less pronounced than with agonist-induced assays. This may be to loss of critical regulators during cell permeabilisation, although it is also possible that there are other functions of iRhoms upstream of IP₃Rs that contribute to the overall effect.

Also as suggested, we measured IP₃R-mediated ER Ca²⁺ release triggered by caged-IP₃. This gave an unexpected result in that both wild type and iRhom1/2 DKO cells released Ca²⁺ after photolysis of caged-IP₃ (new Fig. 3e). At face value, this data contradicts the agonist induced activation and the experiment with permeabilised cells, indicating instead that under

the caged IP₃ conditions IP₃Rs, or at least a subset of the receptor populations in iRhom1/2 DKO cells, can respond to caged-IP₃ stimulation.

Clearly this challenges our conclusion that the reason for the resistance of iRhom1/2 DKO cells to ER stress is a direct effect on IP₃Rs. We do not have a clear explanation for these discrepant results, all of which have been performed multiple times and are fully reproducible. On one side of the argument, we have a number of lines of evidence that support the idea that iRhoms can regulate the activity of IP₃Rs, particular under ER stress. (1) In the absence of iRhoms, we observe reduced ER Ca²⁺ release by GPCR activation by ATP and bradykinin (Fig. 3a-b), and by IP₃ in permeabilised cells (Fig. 3d). (2) There is reduced IP₃R-dependent tunicamycin-induced ER Ca²⁺ release (Fig. 3f) and reduced Ca²⁺ uptake into mitochondria in the absence of iRhoms (Fig. 3h). (3) Tunicamycin-induced apoptosis is inhibited by the IP₃R inhibitor 2-APB (Fig. 3g). (4) iRhom2 binds directly to endogenous IP₃Rs and not to other ER-localised Ca²⁺ channels (Fig. 4b). (5) ER stress increases binding of iRhom2 to IP₃R1 (Fig. 4f). (6) iRhom2 regulates the binding of inhibitory BCL-2 to IP₃R1 (Fig. 6c-d). Finally, (7) tunicamycin-induced apoptosis in iRhom1/2 DKO cells is rescued by removing the inhibition of BCL-2 on IP₃Rs, using BCL-2 RNAi (Fig. 5d) or BIRD-2 peptide (Fig.6a).

Against these conclusions, we set the result that there is no observable difference between WT and DKO cells in response to caged-IP₃. It may be that some aspect of the way in which caged IP₃ is delivered confounds the experiment, or that the acute release by photolysis of the caged IP₃ stimulates a subset of receptors that are iRhom independent, but we have no direct evidence for such an explanation. We therefore think it is more rigorous at this stage to describe these data and be explicit about their divergence.

We stress that this unexpected result does not challenge our major conclusions that iRhoms regulate ER-induced cell death by inhibiting the calcium-mediated mitochondrial apoptosis pathway; nor that this calcium signalling depends on IP₃ receptors; nor that the mechanism relies on modulating the inhibitory effect of BCL-2.

We have carefully described these experiments and our conclusions in the revised text, including adding a paragraph to the Discussion.

7) 2-APB is not a specific inhibitor of IP₃Rs and has major effects on Ca²⁺ entry channels. I think the authors should mention this as a caveat in the description of Figure 3f.

Agreed. We now mention the possible non-specific effect of 2-APB in the results.

8) Why is there so much cleavage of the mutant construct used in Fig 4e? The decreased binding to IP₃R1 may be due to inefficient targeting of the mutant to the ER. The same problem may apply to the interpretation of the data with N-terminal mutants in Fig 5f.

iRhom blots are never very simple. It is well established that iRhoms express as full-length protein at around 100kDa, and a cleaved form at around 50kDa when tagged at the C-terminal. Depending on the type of expression (endogenous, stable, transient), extra bands are often observed, likely due to further proteolytic cleavage of the full-length protein. In this case, even though the mutant iR2_TMD1_IRHD undergoes more cleavage, it is expressed at comparable level of the wild type protein. We have also previously shown by immunofluorescence that it is targeted to the ER like wild-type iRhom2 (Suppl. Fig. 4c). We have now added data that the mutant construct behaves like wild-type iRhom2 in its interaction with, and ability to degrade, EGF (Suppl. Fig. 4d), further supporting the conclusion that it is not substantially mistargeted. But overall, we acknowledge the important caveats associated with this approach and have made them clear in the text.

As suggested, we have now also included immunofluorescent images of the iRhom2 N-terminal mutants used in Fig. 5f to confirm their correct targeting to the ER as wild type iRhom2 (Suppl. Fig. 5).

9) One feature of this study is that different end-points are used for ER stress even in the same series of experiments. For example caspase-3 effects are shown in Fig5b and caspase-9 effects are shown in Figure 5d. Is there a reason for this?

Point well taken about the inadvertent lack of consistency in the end-points shown. We have adjusted the figures to use the same readouts in all cases.

10) In Figure 5d the authors conclude that knockdown of BCL-2 abolished the suppression of tunicamycin ER stress in iRhom2 KO cells based on the intensity of the caspase-9 cleaved band. However, the intensity of this band is dependent on the abundance of its precursor which appears higher in the Bcl-2 KO cells. In any case, some kind of quantitation which takes this into account is necessary to back up this conclusion.

As indicated in our response to the previous comment, we have replaced the figure with caspase-3 as an end-point throughout Fig. 5d and also included a quantification of the blot.

11) A key part of the model shown in Fig 6e is that ER stress dissociates the iRhom/BCL-2 complex from IP₃R and permits mitochondrial Ca²⁺ overload. However, the paper has no experiments (or speculations) on why this dissociation occurs.

As discussed elsewhere, we do not claim to fully reveal the molecular mechanism by which iRhom derepresses BCL-2 inhibition of IP₃R signalling (most obviously under ER stress, but observable also under basal conditions). Although we can speculate a little in the discussion (and have amended it accordingly), our preference is to be open about the limitations as well as the significance of our work. We do agree that the model figure was not well designed. We have now amended it with the aim of more simply and accurately representing our data. We have also discussed whether the model is in fact necessary, considering the possibility that it is more confusing than clarifying. On balance, we think that the revised version does help to summarise the novel conclusions of our work, without over-complication or suggesting that we know more than we do. Of course, we would be happy to take editorial and reviewer advice on the use of this figure panel.

REVIEWERS' COMMENTS

Reviewer #1 (Remarks to the Author):

The authors have responded to all of my concerns/suggestions in a positive manner. This manuscript is now acceptable for publication in Nature Communications.

Reviewer #2 (Remarks to the Author):

The authors have done a lot of work to revise and improve. There are just two minor points that the authors may wish to address.

Regarding time course for death with tunicamycin that we requested, only includes one additional time point at 36h. However, the result is still consistent with the iRhom KO cells being more death resistant.

The reconstitution with iRhom2 in Figure S4e does not seem to facilitate ADAM17 processing (despite efficient co-IP), as shown in top ADAM17/ConA blot, lane 5. If the authors could comment why there is not more of the 80kD processed band, and that the expressed form of iRhom is not behaving as efficiently as endogenous, this will suffice.

Reviewer #3 (Remarks to the Author):

The authors have adequately addressed my previous concerns through providing new data and textual changes. As such, the manuscript is now substantial stronger and more convincing.

Reviewer #4 (Remarks to the Author):

The authors have made a good effort to address my comments. However, the new experiments in which they examined IP3R-mediated Ca²⁺ release directly in permeabilized cells and after uncaging IP3 raise concerns regarding interpretation of the data. In the new Figure 3D the dose-response curve shows no differences between WT and iR1/2 DKO cells up to 1 μ M [IP3]. Only 3 of the data points show differences and at the maximal [IP3] there is only an 18% inhibition. This can be contrasted with the huge effects on the cytosolic Ca²⁺ transients evoked over a full concentration range of ATP or bradykinin. It is unclear what concentration of IP3 was formed in the uncaging experiments but presumably it was a maximal concentration and this produced a statistically significant increase in Ca²⁺ release. I agree with the authors that while this may not make for a neat, tidy, cut-and-dry story a complete and thorough description will provide the basis for future studies that can further clarify the molecular mechanisms by which iRhom proteases regulate intracellular Ca²⁺ release. For completeness, I would suggest that the authors add a sentence on the dose-response relationships shown in Fig 3. In addition, one of the authors in this study (Colin Taylor) is one of several groups pushing the idea that only a small pool of IP3Rs are “licensed” to respond in a cell and has argued for a regulatory role for the actin cytoskeleton and specifically the protein KRAP. Perhaps, iRhoms modify the ‘licensing’ process and these effects would not be visible in the permeabilized cell or uncaging experiments.

Response to reviewers' comments

Reviewer #1 (Remarks to the Author):

The authors have responded to all of my concerns/suggestions in a positive manner. This manuscript is now acceptable for publication in Nature Communications.

Reviewer #2 (Remarks to the Author):

The authors have done a lot of work to revise and improve. There are just two minor points that the authors may wish to address.

Regarding time course for death with tunicamycin that we requested, only includes one additional time point at 36h. However, the result is still consistent with the iRhom KO cells being more death resistant.

The reconstitution with iRhom2 in Figure S4e does not seem to facilitate ADAM17 processing (despite efficient co-IP), as shown in top ADAM17/ConA blot, lane 5. If the authors could comment why there is not more of the 80kD processed band, and that the expressed form of iRhom is not behaving as efficiently as endogenous, this will suffice.

In regard to the first point, we included only one additional time point at 36 h post treatment with tunicamycin as beyond that (for e.g at 48 h), all the control WT cells and most of the iR1/2 DKO cells were dead (likely via both apoptotic and non-apoptotic pathways). Nevertheless, as the reviewer agrees, the results of all earlier time points still clearly show iR1/2 DKO cells are more resistant to ER stress-induced cell death.

For the second point, we believe that the level of mature ADAM17 in cells transfected with iR2 is less than endogenous levels in WT cells, because not all iR1/2 DKO cells will have been efficiently transfected with iR2 construct, and also WT cells do have the additional expression of endogenous iR1, which also regulates ADAM17 maturation.

Reviewer #3 (Remarks to the Author):

The authors have adequately addressed my previous concerns through providing new data and textual changes. As such, the manuscript is now substantial stronger and more convincing.

Reviewer #4 (Remarks to the Author):

The authors have made a good effort to address my comments. However, the new experiments in which they examined IP3R –mediated Ca²⁺ release directly in permeabilized cells and after uncaging IP3 raise concerns regarding interpretation of the data.

In the new Figure 3D the dose-response curve shows no differences between WT and iR1/2 DKO cells up to 1 μ M [IP₃]. Only 3 of the data points show differences and at the maximal [IP₃] there is only an 18% inhibition. This can be contrasted with the huge effects on the cytosolic Ca²⁺ transients evoked over a full concentration range of ATP or bradykinin. It is unclear what concentration of IP₃ was formed in the uncaging experiments but presumably it was a maximal concentration, and this produced a statistically significant increase in Ca²⁺ release.

I agree with the authors that while this may not make for a neat, tidy, cut-and-dry story a complete and thorough description will provide the basis for future studies that can further clarify the molecular mechanisms by which iRhom proteases regulate intracellular Ca²⁺ release. For completeness, I would suggest that the authors add a sentence on the dose-response relationships shown in Fig 3.

In addition, one of the authors in this study (Colin Taylor) is one of several groups pushing the idea that only a small pool of IP₃Rs are “licensed” to respond in a cell and has argued for a regulatory role for the actin cytoskeleton and specifically the protein KRAP. Perhaps, iRhoms modify the ‘licensing’ process and these effects would not be visible in the permeabilized cell or uncaging experiments.

We agree: regulation by iRhoms of intracellular Ca²⁺ release via IP₃Rs appears complex, as highlighted by the results of the caged-IP₃ experiment. As requested, we have now included more description of the dose-response relationship in both the results and discussion sections, to help guide future studies into the relationship between iRhoms and intracellular Ca²⁺ signalling. We have also taken on board and refer to the reviewer’s suggestion about iRhoms potentially having a role in the ‘licensing’ process of IP₃Rs.